



# ISSM-SLPS: geodetically compliant Sea-Level Projection System for the Ice-sheet and Sea-level System Model v4.17

Eric Larour[1], Surendra Adhikari[1], Thomas Frederikse[1], Lambert Caron[1], Benjamin Hamlington[1], Nicole-Jeanne Schlegel[1], Erik Ivins[1], Robert Kopp[2], Mathieu Morlighem[3], and Sophie Nowicki[4]

[1]Jet Propulsion Laboratory - California Institute of technology, 4800 Oak Grove Drive MS 300-323, Pasadena, CA 91109-8099, USA
[2]1 Department of Earth and Planetary Sciences and Institute of Earth, Ocean and Atmospheric Sciences, Rutgers University, New Brunswick, NJ, USA
[3]University of California at Irvine, Department of Earth System Science, Irvine, California, USA
[4]NASA Goddard Space Flight Center, Cryospheric Sciences Lab, Greenbelt, Maryland, USA

**Correspondence:** Eric Larour (eric.larour@jpl.nasa.gov)

**Abstract.**

Understanding future impacts of sea-level rise at the local level is paramount to mitigating its effects. In particular, quantifying the range of sea-level rise outcomes in a probabilistic way enables coastal planners to better adapt strategies, depending on cost and timing. For long-term projections, from present-day to the end of the 21st century, frameworks have been developed

that provide such probabilistic projections. They rely on sea-level fingerprints where contributions from different processes are sampled at each individual time step and summed up to create probability distributions of sea-level rise for each desired location. While advantageous, this method does not readily allow for including new physics developed in forward models of each component. For example, couplings and feedbacks between ice sheets, ocean circulation, and solid-Earth uplift cannot easily be represented in such frameworks. Indeed, the main impediment to inclusion of more forward model physics in prob-

abilistic sea-level frameworks is the availability of dynamically computed sea-level fingerprints that can be directly linked to local mass changes. Here, we demonstrate such an approach within the Ice-Sheet and Sea-level System Model (ISSM), where we develop a probabilistic framework that can readily be coupled to forward process models such as those for ice sheets, glacial-isostatic adjustment , hydrology and ocean circulation, among others. Through large scale uncertainty quantification, we demonstrate how this approach enables inclusion of incremental improvements in all forward models and provides fidelity

to time-correlated processes. The projection system may readily process input and output quantities that are geodetically consistent with space and terrestrial measurement systems. The approach can also account for numerous improvements in our understanding of sea-level processes.





## 1 Introduction

Reliable projections of local sea-level change, together with robust uncertainties, are a key quantity for stakeholders to shape adequate and cost-effective mitigation and adaptation measures to sea-level rise (Kopp et al., 2019). Most regional sea-level projections use a process-based approach, in which all relevant processes are summed up together, including the individual estimates of error, with their spatial signature (Slangen et al., 2012; Church et al., 2013b; Kopp et al., 2014; Jackson and Jevre-
jeva, 2016; Kopp et al., 2017; Jevrejeva et al., 2019). These projections are widely used by coastal planners and stakeholders, as is for example demonstrated by the impact of Kopp et al. (2014, 2017) on assessment reports across the United States (Gornitz et al., 2019; City of Boston, 2016; Kopp et al., 2016; Kaplan et al., 2016; Callahan et al., 2017; Dalton et al., 2017; Griggs et al., 2017; Miller et al., 2018; Boesch et al., 2018).

In their most simple form, these process-based projections (we generally refer to these as KOPP14, in reference to Kopp
et al. (2014)) can be expressed as:

$$RSL_{\text{total}}(\theta, \phi, t) = \sum_{i=1}^{n} F_{\text{GRD},i}(\theta, \phi) \cdot B_i(t) + RSL_{\text{sterodynamic}}(\theta, \phi, t) + RSL_{\text{GIA}}(\theta, \phi, t).$$ (1)

where $RSL_{\text{total}}(\theta, \phi, t)$ is the total projected relative sea-level (RSL) change at time $t$, latitude $\theta$ and longitude $\phi$. For all barystatic processes, or processes that change the total ocean mass, the effects of gravity, rotation and deformation (GRD) on local sea level are computed by multiplying the total barystatic contribution $B_i(t)$ by the associated barystatic-GRD fingerprint
(abbreviated by "fingerprint" from here on), or $F_{\text{GRD},i}(\theta, \phi)$, which is computed a priori. This procedure is generally used to include the effects of glacier (GLA) and ice sheet mass loss, as well as for projected changes in terrestrial water storage (TWS). The effects of sterodynamic sea-level change $RSL_{\text{sterodynamic}}(\theta, \phi, t)$, which is the sum of local thermosteric expansion and local sea-level changes due to ocean dynamics, is generally included by directly using estimates from Earth System Models, such as the output of the Coupled Model Intercomparison Project 5 (CMIP5, Taylor et al., 2009). Finally, the glacial isostatic
adjustment (GIA) term is generally accounted for using output from a periodically updated global model.

To derive uncertainties for these local projections of sea level change, the barystatic components $B_i$ are often sampled from a probability distribution found in published probabilistic projections, for example from expert elicitation projects (e.g. Bamber et al., 2019), or other ice-sheet models (DeConto and Pollard, 2016). The sterodynamic contribution often uses the inter-model spread as a source of the uncertainties. While the basis of each probabilistic projection is similar, each group adds additional
components and physics to Eq. 1. For example, in Kopp et al. (2014) and Kopp et al. (2017), a Gaussian Process Regression model, based on tide-gauge observations is used to account for the effect of non-climatic vertical land motion. Or in Jackson and Jevrejeva (2016) and Kopp et al. (2017), the GRD effects of ocean dynamics (Richter et al., 2013) are explicitly taken into account, with Kopp et al. (2014) computing these effects over the entire projection time series.

One of the key strengths of this approach is how simple and transparent it is, as the process from probabilistic estimates
of the underlying processes into local sea-level changes is a simple multiplication operation with the respective barystatic-GRD fingerprint. It provides a framework that outputs a probability density function (PDF) for $RSL$ at any desired location,



from which the expected sea-level change and its confidence intervals can be derived. This provides both efficient calibration/validation qualities to projections and streamlines incrementally updated projections. In essence, each modular input may be improved separately, so updates are unencumbered by the queueing up of new modules for incorporation into more complex
Earth System Models.

Yet recently, a growing body of research indicates that additional processes should be considered in this process-based approach. Indeed, inclusion of such processes is critical to improving the quantification of uncertainties in local sea-level change predictions, but they are not directly feasible within the framework of Eq. 1. Below, we highlight some of the key contributors to uncertainty that until now, have not been considered together in large-scale estimates of sea-level change.
First, in Eq. 1, the multiplication of a barystatic mass contributor $B_i(t)$ with a fingerprint $F_{\mathrm{GRD},i}(\theta,\phi)$, assumes that the fingerprint stays constant in time, which is not always the case (Mitrovica et al., 2011). Instead, a fingerprint results from feedbacks between the geometry of sea-level components. For example local sea level depends on the geometry of ice mass loss, so temporal changes in ice geometry will directly translate into local sea-level changes (e.g. Larour et al., 2017b; Mitrovica et al., 2018). As a result, this temporal variability not only affects the expected local sea-level changes, but also its uncertainties,
as the uncertainty of the input mass loss also has a profound geometry due to relative limitations in measurement and data interpretation.

Second, covariances in time as well as the co-variances between the individual processes are not always negligibly small, though they are often considered to be or are approximated by a simple relationship (e.g. Church et al., 2013b). Indeed, assuming so could cause a significant misrepresentation of the estimated uncertainties in local sea-level change. For example,
Le Bars (2018) showed that most driving factors of sea level are correlated with global-mean temperature changes, and ignoring this inter-process covariance can underestimate uncertainty in local sea-level change. Note that in addition to co-variances between processes, the uncertainty in individual processes may also be correlated temporally. Propagating this full spatio-temporal covariance into projections and its uncertainties promotes a better understanding of the spatial and temporal coherence of uncertainties, which could, for example, allow us to assess the likelihood of reaching specific sea levels by 2100 given
observed sea-level change during the next 20 years.

Thirdly, recent work on the Antarctic Ice Sheet (AIS) shows a strong coupling between GIA, elastic surface deformation and ice mass loss (Gomez et al., 2018; Barletta et al., 2018; Larour et al., 2019). Such relationships between these processes suggest that any uncertainties in computed ice-sheet histories and solid-Earth properties that propagate into GIA projections (Caron et al., 2018) can also feed back into ice-mass-loss projections, thus considering these processes as independent ignores
these couplings. Here, the main problem is that projection frameworks are articulated in terms of changes in mass, while most ice-sheet models, GIA models, TWS evolution models, and glacier models, are explicitly described in terms of local mass change evolution (or thickness changes, in m/yr water equivalent). In order to be able to account for strong couplings, or to even be able to ingest recent modeling results such as the large set of Modeling Intercomparison Projects (MIPs) results from the CMIP5 or CMIP6 projects (such as ISMIP6 or GlacierMIP), one needs to propagate the local mass changes and the associated
uncertainties into regional sea-level projections.





Similarly, additional strong positive feedbacks between ice sheet and ocean dynamics have been evidenced in work from among others, Goldberg et al. (2012, 2018, 2019) and Seroussi et al. (2017). Specifically, these studies suggest that strong coupling between sub ice-shelf ocean circulation (in particular melt rates) and ice-flow dynamics (in particular, grounding line dynamics and mass transport resulting in modifications of an ice-shelf draft) results in significant retreat of ice streams such

as Thwaites Glacier and Pine Island Glacier, as Antarctica's warm circumpolar deepwater is advected close to their grounding line. Other higher-frequency processes such as ocean tides and in particular how tidal currents affect water mass properties at ice sheet marine margins (Padman et al., 2018) are critical in understanding how mass loss rates will evolve. This will significantly impact how melt-rate parameterizations are developed to quantify melt rates, especially in the West Antarctic Ice Sheeet area (Seroussi et al., 2017). Significant work remains in calibrating such melt rate parameterizations to correctly account

for all afore-mentioned effects. While more work is required in terms of constraining such parameterizations, the impact of such ice/ocean feedbacks have not been assessed in probabilistic sea-level models (PSLMs).

Finally, in the past decade, extensive work has been carried out to probabilistically characterize components like GIA (Whitehouse et al., 2012; Gunter et al., 2014; Caron et al., 2018; Melini and Spada, 2019) or ice-sheet mass balance (Larour et al., 2012b, a; Schlegel et al., 2013, 2015, 2016, 2018). Substantial understanding of the impact of rheological parameters and ice

history on the distribution of bedrock uplift and rate of change in geoid rates has been generated through modelling of GIA. Similarly, for ice sheet models, significant knowledge has been generated about how the mass balances of both AIS and Greenland Ice Sheet (GIS) are impacted by surface mass balance (SMB), ice shelf basal melt, ice/bedrock friction, geothermal heat flux, or ice rheology (see e.g. Fig. 1a).

Moving from strategies where continental scale mass changes are sampled and multiplied with the corresponding fingerprint,

to actually sampling upstream model inputs, such as SMB, basal friction or ice and solid-Earth rheological properties among others, with consideration to spatial patterns of mass change and its uncertainty (see e.g. Fig. 1b-d), is paramount to improving the state of the art. For example, Eq. 1 relies on masses that are agregated at the basin/continental level. However, most ice sheet models compute high-resolution thickness change patterns that are not agregated. This agregation greatly reduces the complexity in representation of model physics and uncetainty propagated at the interface between ice-sheet models and PSLMs.

A more comprehensive approach that reestablishes interfaces between forward models and PSLMs is therefore necessary, where model outputs are not agregated or simplified.

Here we propose a new framework for sea-level projections that is able to account for all terms in Eq. 1. We improve the existing process-based approach by using the Ice-Sheet and Sea-Level System Model (ISSM, Larour et al., 2012c) which allows for inclusion of forward model physics, retaining the covariance between input processes from Eq. 1 and through computation

of high-resolution barystatic-GRD patterns for each individual sample.



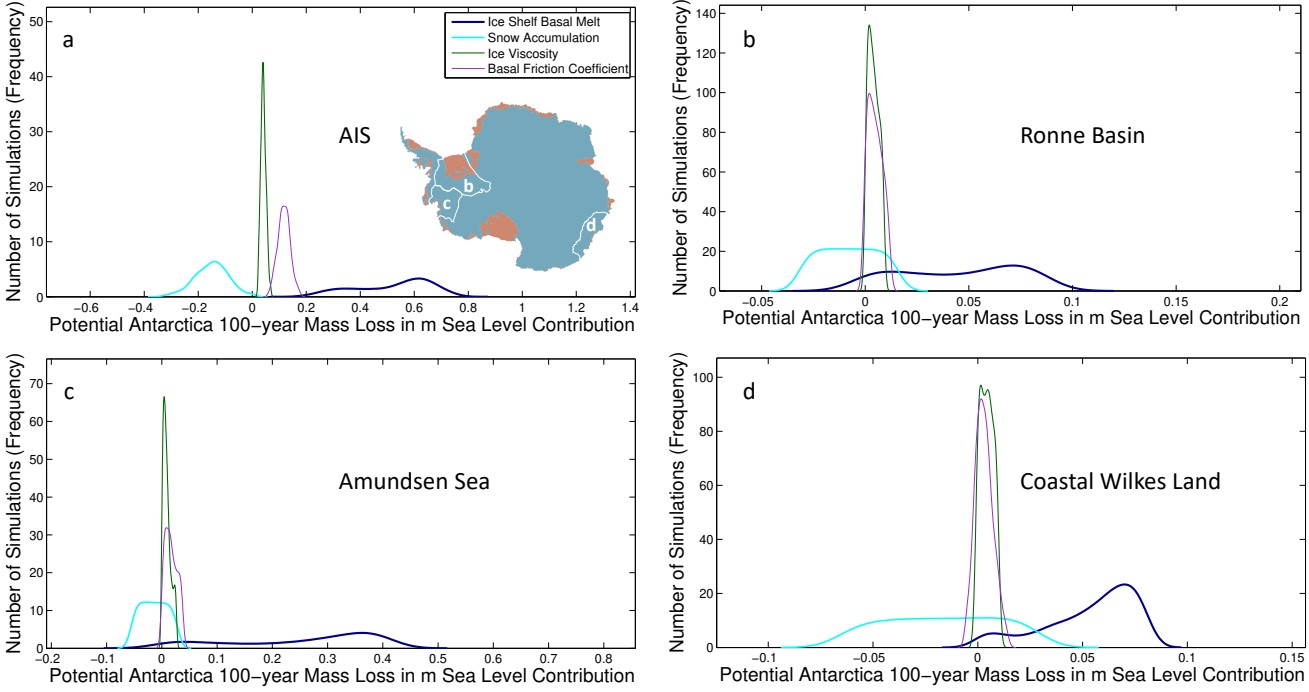

**Figure 1.** Contribution to uncertainty in 100-year extreme warming simulations of AIS and three subregions of the AIS, tested for four different model variables independently. Each probability distribution function represents an ensemble of 800 Ice-sheet and Sea-level System Model ice dynamics (ISSM-ICE) runs, sampled using the ISSM-DAKOTA uncertainty quantification framework. (Schlegel et al., 2018)

## 2 Methods

### 2.1 Theory

Sterodynamic sea-level changes form a significant contributor to both global-mean sea-level rise and are responsible for large parts of the regional deviations from the global-mean projected changes (e.g. Slangen et al., 2012; Church et al., 2013b;

Slangen et al., 2017). Following the CMIP5 conventions, sterodynamic sea-level changes consist of a global-mean thermosteric contribution (variable name `zostoga`) and a local dynamic contribution (variable name `zos`) with a zero mean over the oceans. Generally, an ensemble of model runs, either based on multiple models (e.g. Church et al., 2013b) or on large-ensemble experiments based on perturbing a single model (for example Little et al., 2017), can be used to directly sample regional sea-level changes. An alternative approach to generate more samples than model ensemble members is to determine common

modes of variability, for example by extracting the largest empirical orthogonal functions from each model and perturbing the associated principal components.





While sterodynamic effects do not change the total ocean mass, ocean dynamics can drive redistribution of ocean mass, which causes ocean-bottom pressure changes, particularly on shallow shelf seas (Landerer et al., 2007). These bottom pressure changes cause a change in the load of the solid Earth below, and thus result in GRD effects, which are often referred to as self-

attraction and loading (SAL) effects. These SAL effects could cause several centimeters of additional sea-level rise above the sterodynamic signal in century-scale sea-level projections (Richter et al., 2013). By adding the ocean-bottom pressure changes to the sea-level equation solver, this effect can be incorporated in regional sea-level projections.

As depicted in Eq. 1, in the classical approach, static sea-level fingerprints are computed a priori for each individual process, which typically include glaciers (GLA), the Greenland and Antarctic Ice Sheets and terrestrial water storage (TWS). These fin-

gerprints are subsequently multiplied by the equivalent barystatic contribution, which is often sampled from a PDF, and added, together with the sterodynamic and GIA contribution to obtain local RSL changes and the associated confidence intervals. This method is both transparent and simple, while maintaining computational efficiency owing to the fact that the fingerprints are do not have to be computed for each sample or time step.

However, several issues arise from this approach, which can be mitigated using a different method. First, it is assumed that

the spatial pattern of mass loss is known a priori and does not vary over time. A common approach is to assume that the mass loss is uniformly distributed over the ice sheet, or that it follows the spatial pattern derived over the GRACE period. Jackson and Jevrejeva (2016) quantified the errors induced by assuming a uniform mass loss, and found that this bias could be up to 1 cm and $\geq$ 10 cm for sites distant from and close to centers of mass loss. Furthermore, the approximation of time-invariant fingerprints could lead to biases, when the spatial pattern of mass loss varies over time.

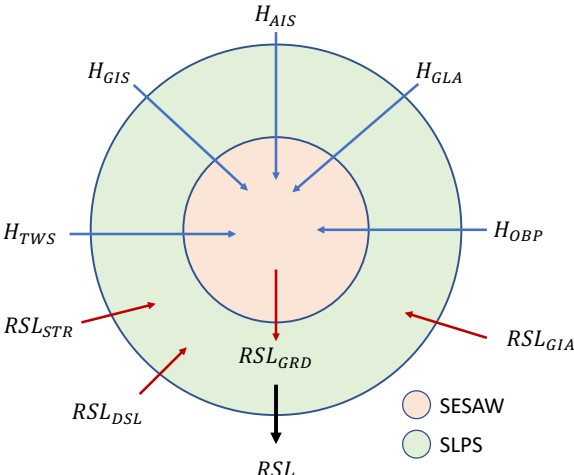

**Figure 2.** Diagram ISSM's Sea-Level Projection System (ISSM-SLPS) model. The system is driven by requirements from Eq. 2. ISSM-SESAW is the GRD core of the system ( in pink). ISSM-SLPS is a combination of ISSM-SESAW and a layer (in green) that handles STR, DSL and GIA inputs, as well as all uncertainty quantification aspects.





In our approach (Figure 2) ISSM Sea-Level Projection System (ISSM-SLPS) solves for RSL as follows:

$$RSL(\theta,\phi,t) = RSL_{STR}(t) + RSL_{DSL}(\theta,\phi,t) + RSL_{GIA}(\theta,\phi,t) + RSL_{GRD}(\theta,\phi,t). \quad (2)$$

The first two terms on the right hand side, i.e. $RSL_{STR}(t) + RSL_{DSL}(\theta,\phi,t)$, together represent the sterodynamic sea level change. $STR$ represents the local thermosteric expansion and $DSL$ local sea-level changes due to ocean dynamics. These can be obtained from CMIP results. The GIA contribution to ongoing sea level change, $RSL_{GIA}$, is given for example by Caron

et al. (2018). The last term, $RSL_{GRD}$ refers to the component of sea level change due to mass induced GRD response of the solid Earth (Gregory et al., 2019), excluding the GIA processes. This includes mass transport between the land and the ocean, as well as that due to dynamic ocean circulation. The latter field is provided by CMIP as the ocean bottom pressure (OBP) products. Note that GRD associated with land-ocean mass transport is usually termed "sea level fingerprint" (e.g., Mitrovica et al., 2009), while the GRD due to OBP variability is termed "self-attraction and loading" phenomenon (e.g., Ray, 1998). As

we shall see, we unify both of these elements of contemporary GRD sea level in equation (3).

We compute $RSL_{GRD}$ using ISSM's Solid Earth and Sea-Level Adjustment module (ISSM-SESAW; Adhikari et al., 2016a). Assuming that all of land ice/water mass change directly modulates the ocean mass, we define a global mass conserving loading function, $M_{global}(\theta,\phi,t)$, that describes the change in mass per unit area on the solid Earth surface as follows:

$$M_{global}(\theta,\phi,t) = M_{land}(\theta,\phi,t)\Big[1 - \mathcal{O}(\theta,\phi)\Big] + \rho_o\Big[H_{OBP}(\theta,\phi,t) + RSL_{GRD}(\theta,\phi,t)\Big]\mathcal{O}(\theta,\phi), \quad (3)$$

where the land loading function (with dimensions of mass per unit area) $M_{land}(\theta,\phi,t)$ is given by Adhikari et al. (2016b):

$$M_{land}(\theta,\phi,t) = \rho_i\Big[H_{AIS}(\theta,\phi,t) + H_{GIS}(\theta,\phi,t) + H_{GLA}(\theta,\phi,t)\Big] + \rho_w H_{TWS}(\theta,\phi,t). \quad (4)$$

Here, $\rho_i$ is the ice density, $\rho_w$ is the freshwater density, $\rho_o$ is the mean density of ocean water, and $H_{OBP}$ is the (ocean) water equivalent height of the ocean bottom pressure change. Similarly, $H_{AIS}$, $H_{GIS}$, and $H_{GLA}$ are the ice height change in the respective cryospheric domains, and $H_{TWS}$ is the freshwater height change in the non-cryospheric land domain. Note that

we invoke an ocean function $\mathcal{O}(\theta,\phi)$ in equation (3) to ensure mass conservation in the system.

The contemporary mass transport function $M_{global}(\theta,\phi,t)$ loads the underlying solid Earth that is self-gravitating, rotating, and viscoelastically compressible. The induced spatial pattern of $RSL_{GRD}(\theta,\phi,t)$ is dictated by the perturbation in Earth's gravitational and rotational potentials and associated viscoelastic deformation of the solid Earth (Farrell and Clark, 1976; Milne and Mitrovica, 1998). In the absence of dynamic sea level and meteorologically induced high-frequency signals, the sea surface

height mimics the spatial pattern of geoid (Gregory et al., 2019). Therefore, we may write

$$RSL_{GRD}(\theta,\phi,t) = C(t) + G_{GRD}(\theta,\phi,t) - B_{GRD}(\theta,\phi,t), \quad (5)$$

where $G_{GRD}(\theta,\phi,t)$ and $B_{GRD}(\theta,\phi,t)$ represent the change in geoid and bedrock elevation induced by the loading of the solid Earth (equation 3), respectively. Spatial invariant $C(t)$ is invoked to ensure mass conservation in the Earth system, and it may be readily derived by inserting equation (5) into equation (3) and integrating it over the solid Earth surface.





Both $G_{GRD}(\theta,\phi,t)$ and $B_{GRD}(\theta,\phi,t)$ appearing in equation (5) may be partitioned into two components each: those re-
lated to gravitational potential and those to rotational potential. The former components can be computed by convolving
$M_{global}(\theta,\phi,t)$ with respective Green's functions. These may be defined in terms of surface harmonics with loading Love
numbers as coefficients. Given the structure and viscoelastic properties of the solid Earth, these numbers characterize the ax-
isymmetric deformational and gravitational response of Earth to the applied unit surface load. The rotational components de-

pend upon tesseral second-degree loading and tidal Love numbers as well as on the perturbation in Earth's inertia tensor, which
in turn depends on $M_{global}(\theta,\phi,t)$. In order to solve for $R_{GRD}(\theta,\phi,t)$, we require an a priori knowledge of $M_{global}(\theta,\phi,t)$,
which in turn depends on $R_{GRD}(\theta,\phi,t)$ itself. The system of equations (3) and (5) is therefore solved iteratively until a desired
solution accuracy is achieved. One key feature of this field is that as ice sheets lose mass, the near-field relative sea level drops,
and far-field sea level rises at a much larger rate than the barystatic term for the sake of mass conservation. While theoreti-

cal/numerical treatments on the topic are found elsewhere (e.g., Farrell and Clark, 1976; Mitrovica and Peltier, 1991; Mitrovica
and Milne, 2003; Spada and Stocchi, 2007; Adhikari et al., 2019), version 1.0 of the SESAW algorithm where $RSL_{GRD}$ is
solved for is presented in Adhikari et al. (2016a).

## 2.2   Meshing

SESAW is a mesh based convolution based on Eq. 2 in Farrell and Clark (1976). As such, it relies on an anisotropic unstructured

mesh of the surface of the Earth which is refined according to specific metrics such as distance to the nearest coastline, presence
of loads (such as changes in ice thickness or TWS), and the complexity of the coastline. Given the amount of inputs being
sampled for in the SLPS system, a systematic approach to refining such a mesh needs to be developed. The main tool for
such a refinement is the ISSM implementation of the BAMG anisotropic mesh refiner (Hecht, 2006a, b). This is a 2D based
anisotropic mesher which can refine a mesh according to several constraints at the same time: a metric to specify directions

along which the mesh resolution needs to be improved, specific vertex or segment positions, in particular vertex positions of
the region outlines, and specified mesh resolutions for user-defined locations. Combining these constraints, we develop an
approach based on meshing of a set of 2D continental areas of the Earth, projection of such 2D meshes onto the 3D Earth
surface and then stitching of the resulting meshes into one seamless global 3D mesh.

A plot of the 2D regions is given in Fig. 3, which include South America, North America, Australia, Eurasia and the Pacific

regions. At the North and South, we have regions defined for Antarctica, and Greenland. Greenland itself has been further
refined into 18 regions drawn along the main ice divides of Greenland, following (Zwally et al., 2012, Fig.3). The approach
facilitates a direct linkage of models from the existing literature, or potentially from previous ISSM studies such as Seroussi
et al. (2017); Schlegel et al. (2018), without having to remesh the entire Earth. This in turn allows for direct comparisons
between uncertainty quantification projection results where only one specific region is modified, hence allowing an approach

where control runs can be compared against specific variations of an uncertainty quantification projection run.

An example mesh of the South-American continent is shown in Fig.4. This mesh relies on defined vertices for the outline,
which match the outline vertices for the Pacific, Antarctica, and Eurasian meshes, so that the stitching within a larger 3D mesh
can be done without redundancy in vertices along continental boundaries. In addition, GRACE ice mass trends from 2003 to

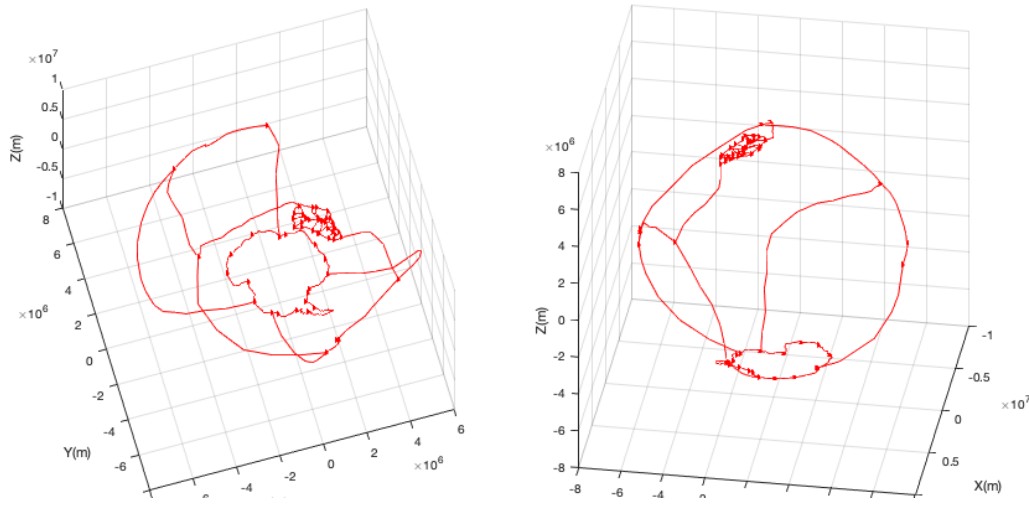

**Figure 3.** 3D plot of the boundaries used to mesh each continental area of the Earth surface. Regions include South and North America, Australia, Eurasia and the Pacific regions, as well as Greenland and Antarctica. In thsi particular scenario, Greenland has been sub-dividied into 18 regions along the boundaries defined in Zwally et al. (2012).

2016 (Adhikari and Ivins, 2016) are provided as a metric to be used for refinemenent of the mesh, in particular around the
Patagonian ice fields. The minimum mesh resolution attained for this mesh is 500 m, and the largest is 1,400 km. Finally, the Global Self-consistent, Hierarchical, High-resolution Geography Database $GSHHS\_c\_L1$ (Wessel and Smith, 1996) was used as a vertex constraint, so that the final mesh perfectly coincides with the coastline dataset (in black). This allows for the most optimum sea-level solution using the SESAW solver.

Once each region has been meshed in 2D using BAMG, it is projected onto latitude and longitude, and concatenated together
to create a 3D mesh. This is possible because each 2D mesh relies on the same set of boundaries as shown in Fig. 3. The resulting mesh is shown in Fig. 5, and comprises 38944 surface elements for 19486 vertices.

### 2.3 Sampling and partitioning

In order to sample variables at each time step, our approach is to use a geographical partitioning of the unstructured mesh. An example is shown in Fig. 6, where a range of values from 1 to 5 has been attributed for each vertex (and element) of the mesh,
corresponding respectively to Antarctica (1), Greenland (2), Glaciers (3), the ocean (4) and land (5). For each partition and for each variable that is probabilistically sampled, we define a probability density function (PDF). For normal distributions for example, this will be done through a mean and standard deviation.





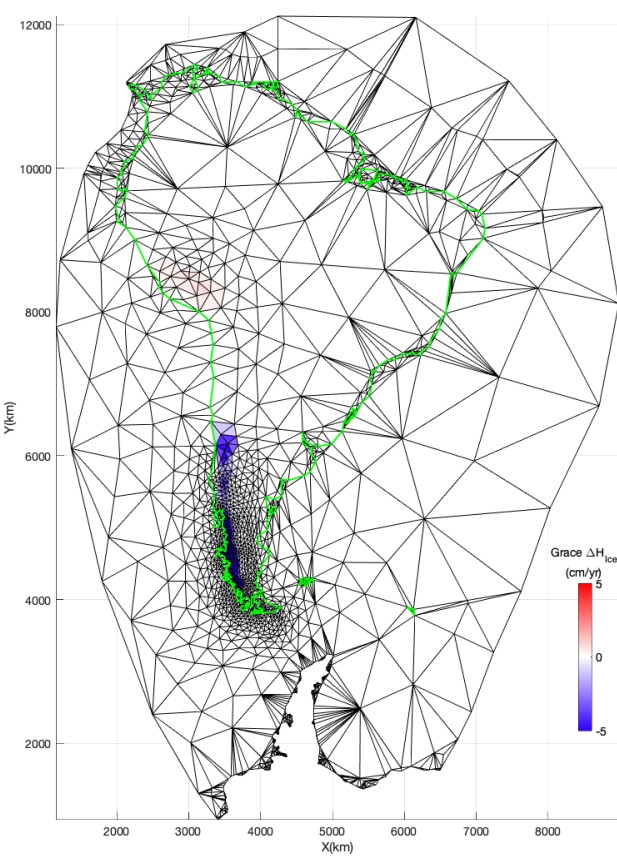

**Figure 4.** 2D Adaptive mesh of South-America using GRACE observations of ice mass change (in cm/yr) from 2003 to 2016. Seismic effects (Richter et al., 2019) are not removed in this rendering of Patagonian ice mass loss that was directly taken from Adhikari and Ivins (2016). The Global Self-consistent, Hierarchical, High-resolution Geography Database (GSHHGD) coarse L1 coastline is shown in green. Segments of the triangular mesh are plotted in black.

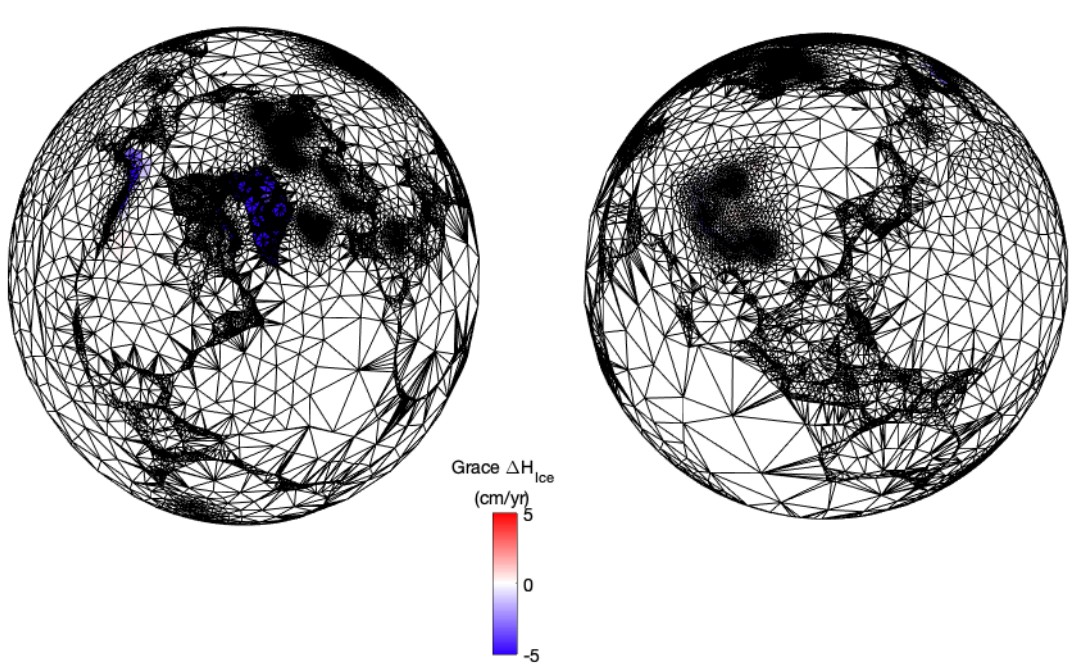

**Figure 5.** 3D Earth mesh stitched from 3D projections of 2D regional meshes of the following regions: South and North America, Australia, Eurasia, the Pacific regions as well as Antarctica and Greenland. GRACE observations of ice mass change (in cm/yr) from 2003 to 2016 (Adhikari and Ivins, 2016) are overlayed over the mesh. Left frame azimuth is 30° with elevation of 64°. Right frame azimuth is 205° with elevation of 23°.





The algorithm for sampling through SLPS is explained below, in the generic case where spatial covariances are available between variables.

```
for t=2019:2100,
        pdf=PDF(type, pdfspec_arg,t);
        for j=1:nsamples,
            alphas = DAKOTA->sample(pdf,j); %alphas array of size [NUM(VAR),RANGE(partition)]
            for VAR in  (DSL,STR,H_AIS,H_GIS,H_GLA,H_TWS):
230             for VERTEX in  MESH:
                    for i=1:range(PARTITION):
                        if PARTITION(VERTEX)==i,
                            alpha=alphas(VAR,i)
                            VAR(VERTEX)=VAR0(VERTEX)*alpha;
235                     end
                    end
                end
            end
            RSL(j,t)=SLPS(DSL,STR,H_AIS,H_GIS,H_GLA,H_TWS);
end
    end
```

where t is the time variable (ranging from year 2019 to 2100, at 1 year intervals), j is the counter for each sample, from 1 to nsamples (in our case, 10,000), VAR is the sampled variable (from one of the SESAW inputs, excluding $RSL_{GIA}$ which is deterministic in our framework), VERTEX is a counter for all vertices in the mesh MESH, PARTITION is the partition vector (for example ranging from 1 to 5 in Fig. 6), PDF is the joint probability distribution of variables across all geographical locations, DAKOTA is the sample generator in ISSM (Eldred et al., 2008; Larour et al., 2012b), alphas the $j^{th}$ sample matrix of scaling factors with size (number of variables, number of partitions) , VAR0 the unmodified variable (stored in memory at the beginning of the model run), SLPS is the sea-level solver, generating $RSL$ for a specific sample of all the probabilistic variables. In this algorithm, the PDF distribution is built by specifying its nature and parameters, e.g. the 'type' argument can indicate the choice of a multivariate Gaussian distribution and 'pdfspec_arg' specify the vector of means and covariance matrix of alpha between each other and between partitions.

For this application, we assume that each variable and each partition is independent, and we set the mean of all distributions to 1. This ensures that values of $alpha$ behave as scaling parameters. We use them to directly scale a variable locally, according to which partition area this location geographically belongs. This method is therefore significantly different from the appraoch in KOPP14, where the entire mass within a certain partition (for example GIS or AIS) is sampled. Here the sampling is a scaling of a vectorial field, which therefore preserves the local geographical distribution of a given variable. This is shown in Fig. 7 for a scenario where thinning rates of the GIS are sampled using one geographical partition (corresponding to Fig. 6 partition value of 2, in blue). We display for a standard deviation $\sigma = 5\%$ the value of the thinning rate samples at three standard deviations from the average. The structure of the thinning rate as it is sampled is kept intact, implying that the spatial covariance of the variable being sampled across the mesh is kept closely similar across samples and within any given partition.

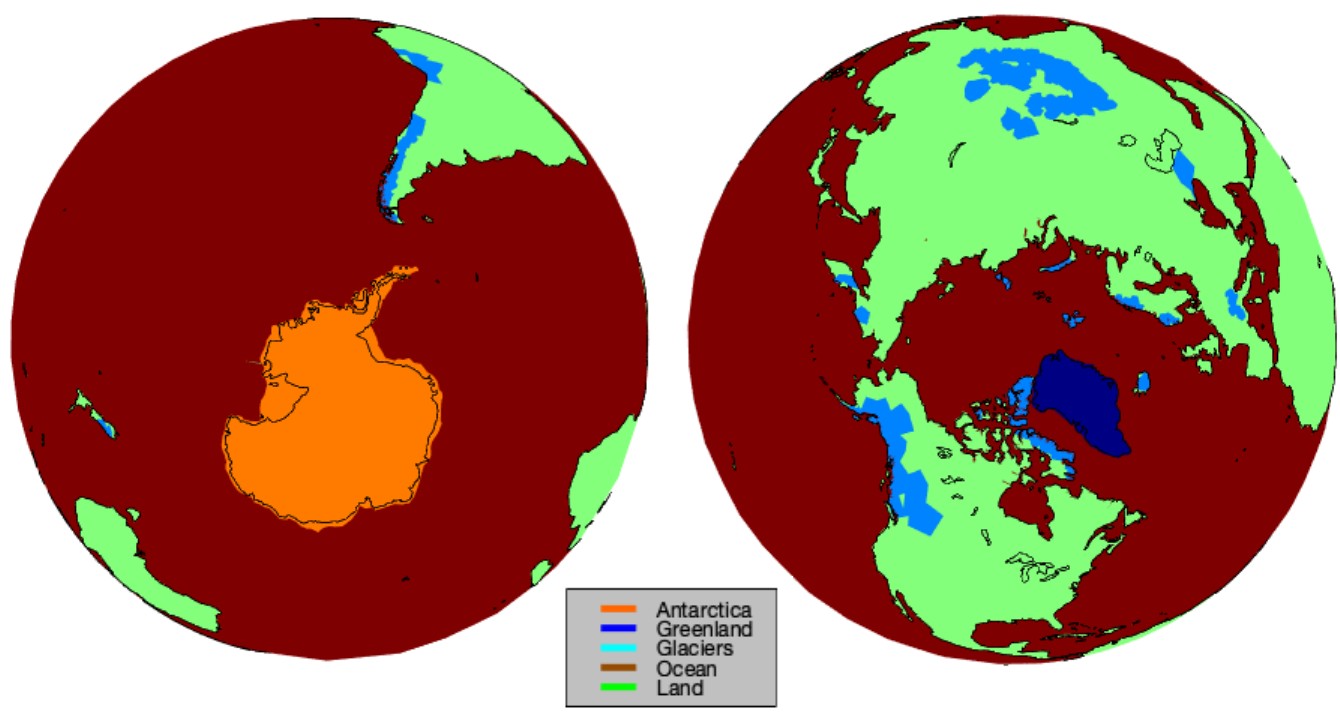

**Figure 6.** Partition vector (values from 1 to 5, 1 for Antarctica, 2 for Greenland, 3 for Glaciers around the world, 4 for the ocean and 5 for land excepted glaciers). The partition vector is used to sample probabilistic variables in a geographically consistent way, with PDF distributions moments (mean and standard deviation) defined for each partition area.

## 2.4 Modularity

The advantage of the partition approach as implemented in SLPS is that various approaches to probabilistic projections can be executed with the same framework. First, as we will show in the next section, the KOPP14 approaches are fully compatible with the SLPS framework. Indeed, fingerprint patterns can be recomputed using local thickness change rate patterns that are spatially constant on the basis of only one partition, such as the entire Greenland or Antarctic ice sheet (contrary to Fig. 11 where Greeland is subdivided). Second, existing probabilistic assessments for specific components (such as the impact of changes in surface mass balance or basal friction in Antarctica (Schlegel et al., 2018) on ice thickness changes) can be used directly, using model output (for example for thickness change rates), or PDF distributions from such model outputs. If the uncertainty quantification was done using a Bayesian framework, the model output statistics can be reused directly (using some type of uniform discrete sampling of each model output), hence replicating a Bayesian type exploration approach of SLPS without incurring any additional computational cost (meaning, not having to rerun the analysis carried out to compute such model outputs). Third, ISSM modules can be activated upstream of the SLPS solver, to push further the boundaries of the uncertainty assessment. For example, an analysis of the impact of SMB variations in one specific region of Antarctica could be carried out using the ice-flow modeling core of ISSM, capable of delivering ice thickness changes directly to the SLPS core. Fourth, these modules can be activated while remaining coupled to other modules. For example, in Larour et al. (2019), it was demonstrated that over centennial time scales, coupling between the

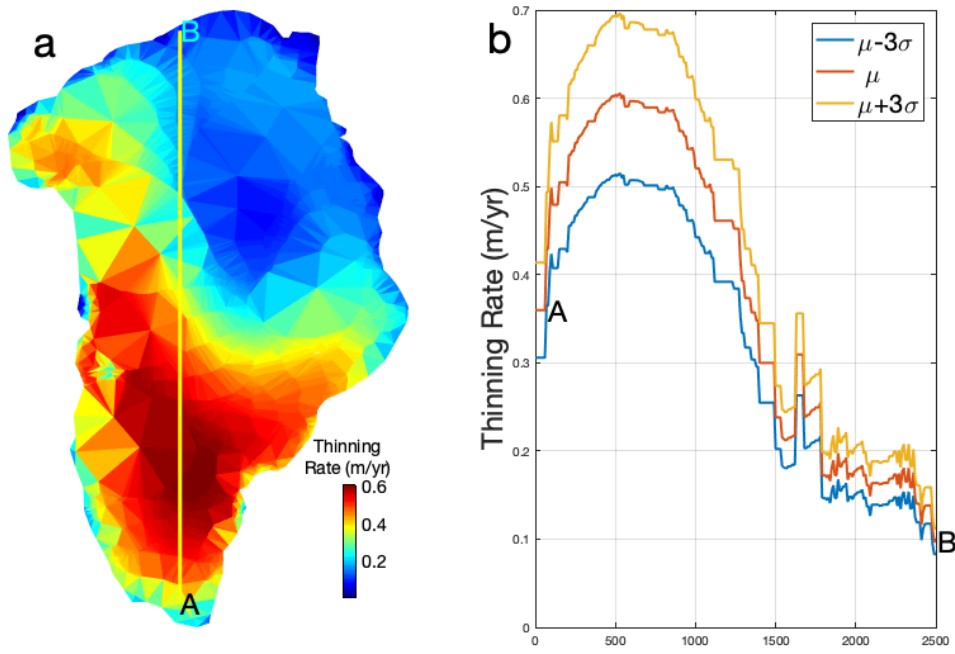

**Figure 7.** Random sampling of thinning rates across Greenland. a) GRACE generated thinning rate pattern at 2005 (in m/yr). b) Thinnning rate along the AB profile (from a) (in red, representing the average of the PDF), and samples generated at $-3\sigma$ (blue) and $+3\sigma$ (yellow) from the average.

elastic uplift of the grounding line, and ice-flow related grouonding line migration, are key to controlling the retreat of Thwaites Glacier

in West Antarctica. Assessing the uncertainty brought by such processes on sea-level rise (SLR) projections would require this coupling to be activated, which could be done (assuming computational costs are still realistic) without modifications to the SLPS framework. Finally, given how closely ISSM can be integrated within Web Server architectures using its native JavaScript interface (Larour et al., 2017a), SLPS is potentially fully compatible with open source types of collaborative approaches where inputs from the community could be provided directly to Web Servers running ISSM in the background, to generate model projections without significant investment in a computation core and/or

an interface to the latter.

## 3 Results and Discussion

SLPS probabilistic projections were validated using model inputs from the Intergovernmental Panel on Climate Change World Fifth Assessment Report (IPCC AR5) (Church et al., 2013a). AR5 supplies several projection components in SLR equivalents: the 'expansion' term ($STR$), the 'glacier' term (which can be converted into an average thickness change rate for $H_{GLA}$), 'antnet' and 'greennet' for net barystatic

contribution from the Antarctica and Greenland ice sheets, which can also be converted into an average change rate for $H_{AIS}$ and $H_{GIS}$ and the 'landwater' term for TWS contribution to SLR (which can be converted into an average change rate for $H_{TWS}$). For each of these



terms, AR5 supplies the mean projection, and the 5-95% percentile confidence interval. We can use this information to calibrate PDF distri-
butions for thickness change rates at each time step, with the mean of each PDF corresponding to the AR5 mean, and the standard-deviation
calibrated from the 5-95% interval (corresponding to the $-1.65\sigma\,to\,1.65\sigma$ interval). Because AR5 does not supply spatial patterns, we rely

on GRACE 2003-2016 thickness change rate patterns from Adhikari and Ivins (2016) for $H_{GLA}$, $H_{AIS}$ and $H_{GIS}$. For $H_{TWS}$, we assume
a uniform spatial distribution over all the spatial partitions. $STR$ is also considered uniform over all the oceans. $DSL$ is not sampled, but
rather deterministically set to the $DSL$ term of the CMIP5 NorESM-ME runs (Bentsen et al., 2013). GIA is independently sampled (from
Caron et al. (2018)) and probabilistically added as an independent PDF. The sampling is carried out on the partitions described in Fig.6 with
the notable exception that the GIS is further divided into 18 different basins as defined in Zwally et al. (2012) and as plotted in Fig.11. For

each year between 2007 and 2100, 10,000 sample runs of SLPS are carried out (with full geodetic capabilities of the SESAW core). For each
partition, samples for the corresponding inputs are generated using a Latin Hypercube Sampling (LHS) algorithm. The runs were carried out
on the Pleiades cluster at the NASA Ames Research Center, on 20 Ivy nodes (20 cores per node for an equivalent 400 cores) over 7 hours.

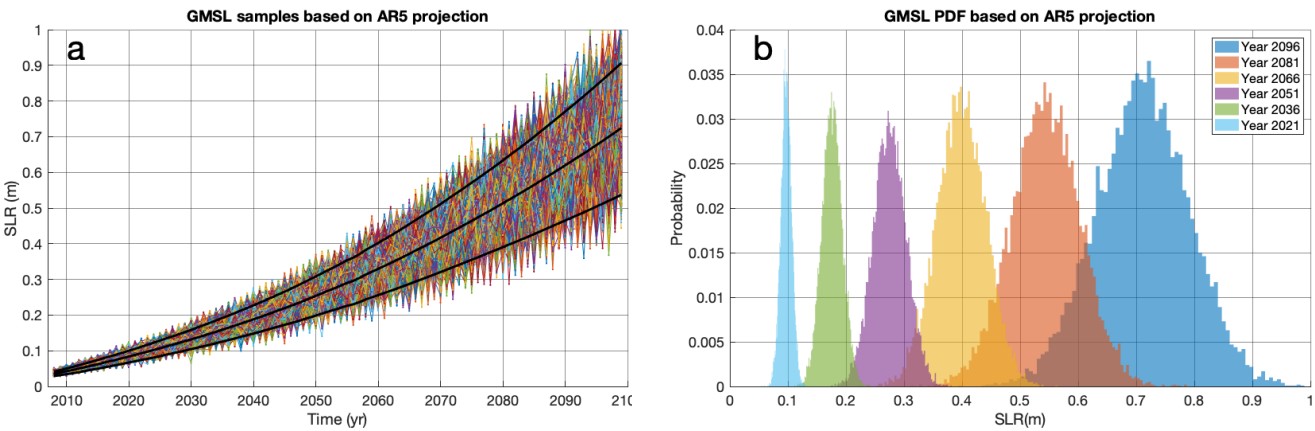

**Figure 8.** ISSM-SLPS projections based on AR5 RCP8.5. For each time step, we sample (10,000 times with Latin Hypercube Sampling, or
LHS) the following inputs: $H_{GIS}$, $H_{AIS}$, thermal expansion of the ocean ($STR$), $H_{TWS}$, and glacier contributions $H_{GLA}$ (see AR5 WG1
Chapter 13, Church et al. (2013b)). Each input's PDF is calibrated using the AR5 5-95% projection confidence interval, similar to Kopp et al.
(2014). The resulting GMSL PDF distribution is shown in a) (in time) and b) (at a sub-set of time steps). The 5-95% confidence interval
(likely range, following AR5 definition) is plotted in black in a), along with the temporal mean. Each time step is fully decorrelated from the
previous time steps, this test being used to validate against existing an existing AR5 projection

Fig. 8 shows projection results for GMSL computed at each time step between 2007 and 2100. We match the mean and 5-95% confidence
intervals of AR5 (Fig.8a) as expected. We also show the evolution of RSL in Fig.8b. for nine cities around the world. We provide the mean

and standard deviation for each PDF, and show how the sampling of ice-related thickness changes impacts mean and standard deviation. In
particular, as expected from the AR5 inputs, we show a marked increase in PDF spreads as time evolves.

Fig. 10 shows the impact of using existing statistics from Caron et al. (2018) to include into SLPS. These statistics were evaluated using
a Bayesian exploration approach, and can be used directly in SLPS, either during a standard probabilistic projection run, or a posteriori as is
the case here. The impact of the migrating Laurentide isostatic bulge on Norfolk, Virginia is apparent in Fig.10, with an offset of 16 cm in

the average projection for the city.



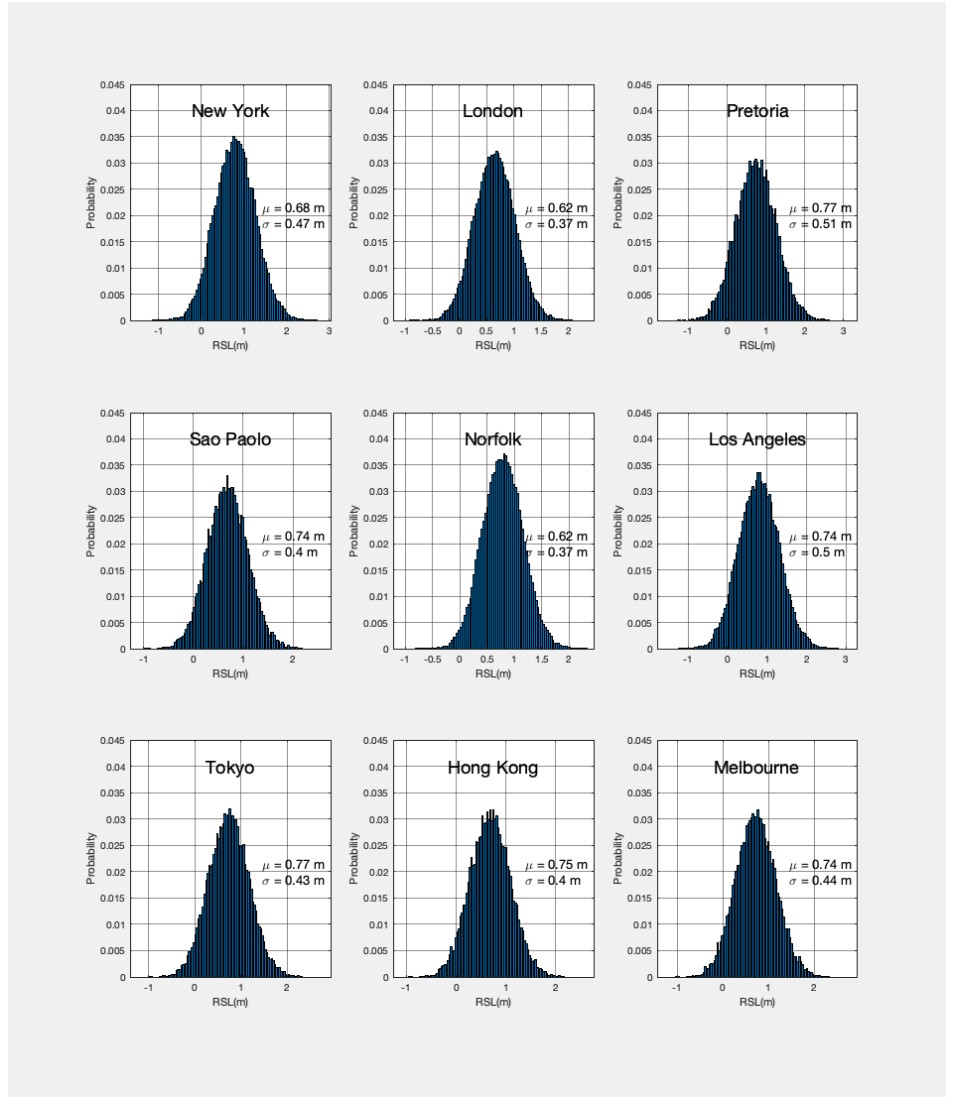

**Figure 9.** AR5 calibrated projection of RSL for nine cities around the world. Sampling was carried out for $H_{AIS}$, $H_{GIS}$, $H_{GLA}$, $STR$ and $H_{TWS}$ using mean and standard deviations from AR5 (Church et al., 2013a). The patterns for ice thickness are from GRACE 2003-2016 trends (Adhikari and Ivins, 2016). $DSL$ is fully deterministic, from the CMIP5 NorESM runs (Bentsen et al., 2013). $RSL_{GIA}$ was deterministically set to 0. Each time step was sampled for using 10,000 LHS samples.

Fig. 11 shows results for a different experiment, in which we quantify the impact of refining the amount of partitions used to sample the uncertainty in ice thickness change rates. For the area of Greenland, we use either one partition (blue boundary), or 18 boundaries (brown basins) from the Zwally et al. (2012) dataset. Each basin is delimited by ice divides, and thus represent a dynamically coherent area, expected to behave (short of ice divides migrating actively) independently from one another. We rerun an SLPS projection using a similar AR5 setup,

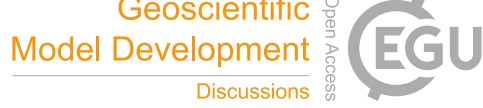



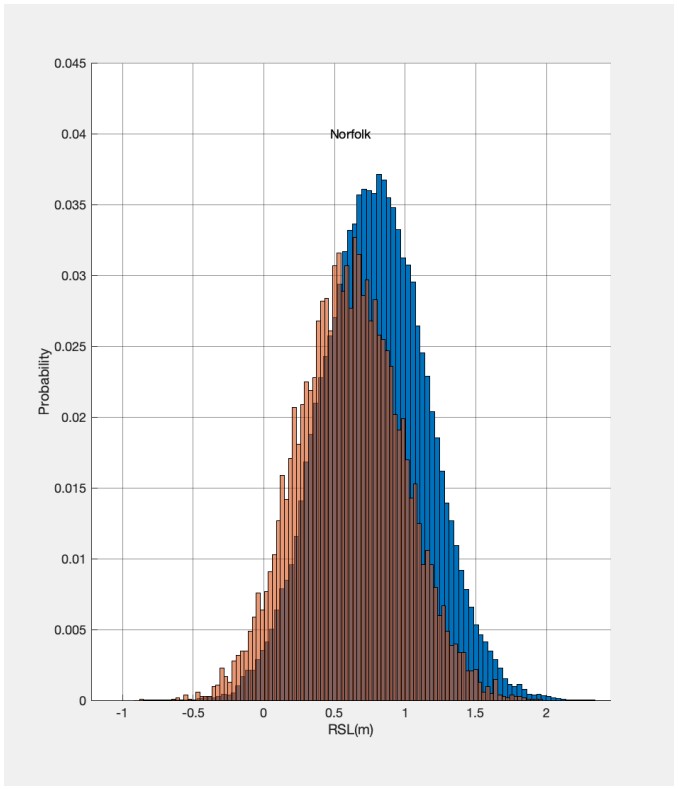

**Figure 10.** AR5 calibrated projection of RSL for Norfolk (in brown) vs same projection in which GIA statistics from Caron et al. (2018) are used to account for GIA induced RSL (in blue)

and display the contribution of ice-related basins to SLR in New York and Hawaii for 1 and 18 partitions respectively. As expected, the mean in PDF distributions are identical for both 1 and 18 partitions. However, the tails are much larger for the 1 basin scenario. The relative difference in standard deviations between 1 and 18 basins ranges from -23% for New York to -34% for Hawaii. This implies that current probabilistic RSL projections are significantly overestimating (by 20-30%) the "likely" (5-95%) range in ice-melt contribution to RSL.

This is understandable because of the fact that in a 1 partition scenario, variations of ice thickness are dictated by scaling of the local ice thickness change rate mean by an identical scalar for the entire partition, which leads to more extreme values for the contribution to RSL. With finer partitions, basins that have low thickness change rates do not inpact RSL as much, and in aggregate the total contribution range varies less. A very similar conclusion was found in Schlegel et al. (2018), where Antarctica had to be subdivided in spatially coherent areas, which were not obvious initially and did not mandatorily map into individual basins. The issue is that the error distribution in model inputs had a specific spatial coherence that had to be respected. Assuming this coherence extended to the entire ice sheet led to significantly larger and unrealistic uncertainty ranges in model outputs. Of course, given differing dynamics in each geographical basin, we cannot assume that the input scaling should be similar (same standard-deviation). This will modify the results in Fig. 11. But our point here is to point out the issue of sub-partitioning as being essential in quantifying the right range of spread in modeled statistical outputs.

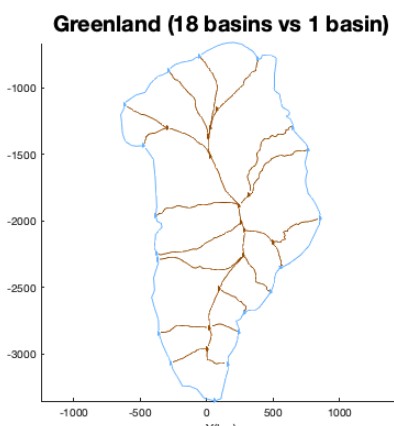
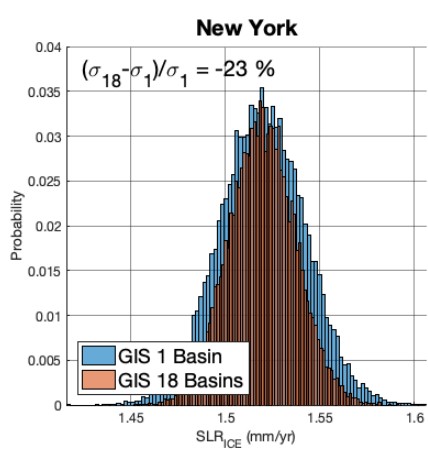
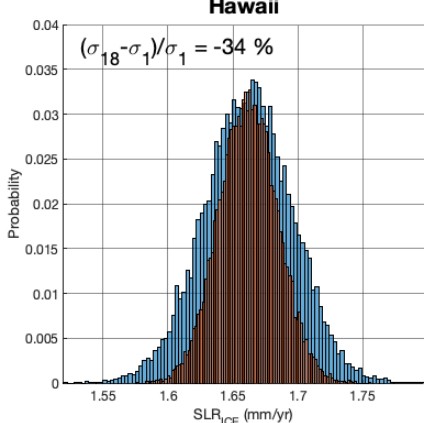

**Figure 11.** Impact of sub-sampling the GIS mass (from 1 basin for the whole ice sheet to 18 basins) on barystatic sea-level rise in New York and Hawaii. The distributions are a result of SLPS, where $H_{GIS}$, $H_{AIS}$ and $H_{GLA}$ were sampled 10,000 times using an LHS algorithm. The mean in PDF distributions for both scenarios are identical, however the tails are much larger for the 1 basin scenario. The relative difference in standard deviations between 1 and 18 basins ranges from -23% for New York to -34% for Hawaii. This implies that current probabilistic RSL projections could significantly overestimate (20-30%) the "likely" (5-95%) range in ice-melt contribution from glaciers and ice sheets.

This analysis also shows that using SLPS, it is possible to efficiently address the question of how to sample uncertainty in a manner that is consistent with the local behaviour of separate basins, glaciers, ice sheets. In Jackson and Jevrejeva (2016) for example, it is shown that the impact of glacier ice thickness variations around the world is siginifically different, and that relying on one fingerprint alone can lead to significant differences in the projection of glacier contribution (up to several percent). Our approach in SLPS ensures that the GRD contribution is systematically reassessed for each sample, at each time step, and the partitioning of our sampling ensures that we correctly capture the specificity of each glacier/ice/hydrological area and their unique mass change trends. It is to be noted that a similar approach is currently implemented in new instantiations of the KOPP14 projection system based on sampling of glacier projections across the 19 Randold Glacier Inventory (RGI) areas used in the GlacierMIP results (Hock et al., 2019). However, these areas can be very large in spatial extent (such as the Low Latitues or North-Asia areas) and should be broken down. Our approach scales for any barystatic contributor, at any spatial scale (example, sub-basin, or at the glacier level) required by the structure of the error distribution of model inputs.

## 4    Conclusions

ISSM SLPS is a new sea-level probabilistic projection system which relies on a new partitioning approach to sampling of boundary conditions, forcings and inputs. It is compatible with previous probabilistic frameworks, but allows for a more robust integration of state-of-the-art results in the modeling of ice flow in ice sheets and glaciers, sterodynamic sea-level, TWS evolution and GIA. It reestablishes temporal correlation in projections where they were previously lacking, and allows for better constraints on spatial and temporal covariances in the model inputs. In particular, it is capable of systematically computing geodetically compliant patterns of sea-level that are consistent with space and terrestial measurement systems. The system relies heavily on the use of high-resolution anisotropic meshes, and allows for a better interfacing with existing modeling frameworks which operate at higher resolutions, and which consistently generate changes in mass density



patterns around the globe. SLPS has been validated against previous frameworks and is fully backwards compatible. Differences between SLPS and previous approaches have also been shown both in terms of integration of GIA statistics, and integration of new high-resolution sampling of ice-thickness change patterns in Greenland. This new approach offers a roadmap towards urther increasing the complexity and realism of sea-level probabilistic projection frameworks.

*Code availability.* The ISSM code and its SLPS components are available at http://issm.jpl.nasa.gov. The instructions for the compilation of ISSM and SLPS modules are available at http://issm.jpl.nasa.gov/download. The public svn repository for the ISSM code can also be found directly at https://issm.ess.uci.edu/svn/issm/issm/trunk, and downloaded using user name "anon" and password "anon". The version of the code for this study, corresponding to ISSM release 4.17, is svn version tag number 24683.

*Data availability.* All datasets used in the projections are freely available in the public domain and are referenced in the text.

*Author contributions.* E. Larour carried out all the simulations and implemented SLPS into ISSM. He wrote the bulk of the manuscript. L. Caron contributed computations for GIA. M. Morlighem contributed enhancements to the adaptive meshers in ISSM. All authors contributed to the manuscript both in terms of text, figures and comments.

*Competing interests.* The authors have no competing interests

*Acknowledgements.* This work was supported by the Jet Propulsion Laboratory, California Institute of Technology under a contract with the
NASA Sea-level Change Team (N-SLCT), NASA Cryospheric Sciences, NASA Modeling Analysis and Prediction (MAP) and NASA Earth Surface and Interior (ESI) Programs as well as the NASA GRACE and GRACE-FO Science Teams Programs. Resources supporting this work were provided by the NASA High-End Computing (HEC) Program through the NASA Advanced Supercomputing (NAS) Division at Ames Research Center.





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
