# Peer review of "ISSM-SLPS: geodetically compliant Sea-Level Projection System for the Ice-sheet and Sea-level System Model v4.17"

_Geoscientific Model Development, 2020_

## Referee Comment (RC1) · Anonymous Referee #1 · 16 Jun 2020

The authors present a framework to produce probabilistic sea-level projections which allows to incorporate and update contributions from different models of the individual drivers in a flexible, yet consistent, way. It builds on the ISSM-SESAW sea level solver by Adhikari et al. (2016), which was also presented in this journal and which is based on an unstructured mesh, allowing high accuracy at very limited computational costs.

The manuscript is clear, it reads well and it presents enough graphics to show the working and capabilities of the software package. Below, I am suggesting a number of minor improvements, listed in the order as they appear in the manuscript.

Line 60: The argument about the time invariancy of fingerprints does not seem very

strong, since it is always possible to increase the number of fingerprints for a given source, with the option of using independent scaling factors for each sub-source. Besides, the authors do not seem to demonstrate the effect of time-variable fingerprints on sea level projections. The improvement could be from a computational point of view (due to, e.g., the modularity of the approach), but then it could be explained more clearly.

Line 65: Not sure what "profound geometry" means, it is possibly a typo.

Line 124: Could you add a reference about the "alternative approach" described here?

Line 148: I guess the term STR refers to a global mean value, not a local one as stated, since RSL_STR only depends on time.

Line 168: I am surprised by the fact that RSL_GRD includes viscoelastic deformation of the solid earth, rather than just elastic as in most studies of this kind. If this is the case, it should be highlighted in the introduction.

Line 215: I suggest adding that the number of elements is significantly smaller than in the case of an equiangular 1x1-deg grid, which would already be rather coarse.

Lines 225-241: I am not sure whether showing a few lines of code is normal in GMD, but, if possible, it would be nice to replace this by a flow chart.

Line 299: It seems a few words are missing, since Fig.8b shows GMSL values, while values for 9 cities around the world are shown in Fig.9. In addition, the caption of Fig.9 does not mention the timeframe for which the projections are generated.

Line 304: I suggest also mentioning that the width of the two PDFs shown in Fig.10 is different, rather than only the mean, because it makes an even stronger point about the benefit of this approach.

Line 316: "impact" (spelling).

Line 338: The importance of "geodetically compliant patterns", in spite of being part

of the manuscript title, is nowhere really discussed nor explained. From this line, my understanding is that it refers to the possibility of defining the mesh in such a way that it matches the location of specific geodetic observations, hence avoiding unnecessary interpolations. In any case, the issue warrants a more extensive discussion, possibly in the introduction.
* * *

---

## Referee Comment (RC2) · Anonymous Referee #2 · 30 Jun 2020

The authors developed a new method to account for feedbacks between ice sheets, ocean circulation and solid-Earth deformation through dynamically computed sea-level fingerprints for probabilistic projections of local sea-level change. This is a significant and timely contribution because current probabilistic projections use static fingerprints and cannot readily incorporate advances in forward models. The manuscript is well-structured and provides clear examples of how the SLPS works. However, I think the clarity of the manuscript could be improved with some (minor) modifications of the text and figures, see the comments below.

L4: For long-term projections (. . .) that provide such probabilistic projections – repetitive, can this be rewritten?

L8: solid-Earth 'uplift' – displacement or deformation would be more complete?

L26: why not give examples of the use of projections globally rather than so many references for just the US?

L37 & 118: sterodynamic sea-level change is defined incorrectly here, see Gregory et al. (2019). It is the sum of global (not local) thermosteric expansion and ocean dynamics (which include the local steric effect, both thermo- and halosteric) including the IB effect. Can the authors clarify this?

L38 & 55: Not just ESMs but also AOGCMs

L47: Do you mean Kopp et al. (2017) here instead of (2014)?

L64-66: Can the authors comment on the expected importance of geometry changes to 21st century sea-level projections?

L83: 'results such as (. . .) results' - please rewrite

L91: 'higher-frequency' higher than what? Perhaps use 'high-frequency' instead?

L97-103: this paragraph misses a final sentence placing its content in context of the manuscript

L104-107: this sentence doesn't read very well, can you please split this up in smaller sentences?

L114-115: 'and through . . . individual sample' misses a verb, perhaps reverse the order with 'retaining the. . .from Eq. 1'

Figure 3: the axis labels have fallen off the figure

Figure 5: It is very hard to distinguish colors of cells within the mesh, could this be improved by for example adapting the range of the colorbar?

L253: has a typo 'appraoch'

L291-292: "DSL is not sampled, but rather deterministically set to the DSL term of the CMIP5 NorESM-ME runs" – why? please explain

L299: refers to figure 9 instead of 8, and discussion of Figure 9 is missing?

L316: "inpact"

L343: 'urther'
* * *

---

## Referee Comment (RC3) · Anonymous Referee #3 · 30 Jun 2020

A review of "ISSM-SLPS: geodetically compliant Sea-Level Projection System for the Ice-sheet and Sea-level System Model v4.17" by Larour et al. for possible publication in Geoscientific Model Development.

The authors present a new geodetically compliant approach for modeling future sea-level rise due to ocean thermal expansion, ocean circulation changes, water mass redistribution, and glacial isostatic adjustment. This new approach has the relative advantage compared to previous approaches (e.g., Kopp et al. 2014, 2017) that coupling and interaction between contributors are taken into account. The authors highlight the important result that, by modeling Greenland ice mass loss using 18 basins rather than

1 basin, uncertainties on future sea-level rise are substantially reduced (Figure 11).

I'll confess that, while I study sea level and was invited to review the paper, I'm a physical oceanographer. I don't have the expertise in modeling, geodesy, or glaciology needed to give a thorough review of this paper. I'd strongly recommend the editor to ensure experts in these topical areas weigh in on this paper.

That being said, I appreciate the paper. I think it's really valuable that the authors are pushing the envelope and developing flexible, modular, coupled approaches to model the various contributors to future sea-level rise and their uncertainties.

I have no major issues with the manuscript (though, again, I strongly recommend more expert reviewers weigh in). I have a couple editorial remarks, detailed below. My only one real complaint regards terminology. The authors' use of gravitational, rotational, and deformational (GRD), sterodynamic, and barystatic sea-level contributions (cf. Equation 1) can be inconsistent with definitions in Gregory et al. (2019). I'd recommend the authors either (1.) adopt the definitions used in Gregory et al. (2019) or (2.) acknowledge where their definitions diverge from Gregory et al. (2019) to avoid confusion.

Specific comments:

Line 2: paramount to -> important for

Line 4: cost and timing -> coast, timing, and risk tolerance

Line 23: are summed -> are modeled separately and summed

Equation 1. Comparing to Figure 3 in Gregory et al. (2019), I'm confused by this equation. The equivalent equation in Gregory et al. would be:

relative sea level = sterodynamic sea level + gravitation, rotation, deformation (GRD) + barystatic - inverted barometer

In Gregory et al. (2019), GRD includes GIA, and GRD makes no contribution to global-

mean sea-level changes. What the authors here call "GRD", Gregory et al. (2019) call "contemporary GRD".

Anyway, it's fine that the authors here use slightly different terminology. But they should acknowledge where their definitions diverge from Gregory et al. (2019). Otherwise, readers (i.e., I) will get confused.

Line 37: local thermosteric -> global-mean thermosteric

Line 53: qualities -> quantities

Line 61: "stays constant in time" it's unclear what the authors mean by this phrase

Line 107: paramount to -> important for

Lines 124-126: I'm unfamiliar with studies doing this for projection purposes. Do the authors have a reference in mind for this technique?

Line 127: can drive redistribution -> can be coupled to redistribution

Line 128: causes -> manifests in

Line 129: cause a change in the load of -> load

Line 130: SAL effects. Suggest to reference, e.g., Ray (1998), Stepanov and Hughes (2004), and/or Vinogradova et al. (2015) on these points.

Line 131: Please add "made by atmosphere-ocean general circulation models (AOGCMs)" after projections and before the Richter et al. (2013) reference

Line 148: local -> global-mean

Line 152-153: The authors should clarify whether they remove the global-mean OBP value or not. If not, are the authors making the Greatbatch correction to account for the Boussinesq nature of most CMIP AOGCMs?

Line 165: The authors should precisely define the ocean function O(theta,phi) for clarity.

Line 176: The former -> These

Line 193: Please define BAMG on first use

Figure 3 caption: thsi -> this

Line 246: alphas -> alpha is

Line 256-257: "We display ... the average" I don't understand this sentence, but maybe it's just me.

Line 262: "the KOPP14 ... SLPS framework" Unclear. Are the authors saying that the approach here reduces to and reproduces the Kopp results under certain strong assumptions? Please clarify.

Line 289: please add spaces between -1.65, sigma, to, 1.65, sigma

Line 291-292: "DSL is not ... CMIP5 NorESM-ME runs" Why only one model and why this model? Variance in model projections of DSL changes can be large and important locally.

Line 303: "Bayesian exploration approach" The authors reference such an approach several times, but never explain it or give a reference.

Line 311: "the tails are much larger for the 1 basin scenario" This seems like a very important results, but I don't think the authors have discussed it enough for me to understand physically why this is the case. Suggest to consider adding more of a description.

Line 311: the "likely" (5-95%) range -> the width of the "likely" (5-95%) range

Line 325: siginificantly -> significantly

Line 343: urther -> further

---

## Author Comment (AC1) · 28 Jul 2020

**1 Response to Anonymous Referee #3**

A review of "ISSM-SLPS: geodetically compliant Sea-Level Projection System for the Ice-sheet and Sea-level System Model v4.17" by Larour et al. for possible publication in Geoscientific Model Development.

The authors present a new geodetically compliant approach for modeling future sea- level rise due to ocean thermal expansion, ocean circulation changes, water mass redistribution, and glacial isostatic adjustment. This new approach has the relative advantage compared to previous approaches (e.g., Kopp et al. 2014, 2017) that coupling and interaction between contributors are taken into account. The authors highlight the important result that, by modeling Greenland ice mass loss using 18 basins rather than 1 basin, uncertainties on future sea-level rise are substantially reduced (Figure 11).

I'll confess that, while I study sea level and was invited to review the paper, I'm a physical oceanographer. I don't have the expertise in modeling, geodesy, or glaciology needed to give a thorough review of this paper. I'd strongly recommend the editor to ensure experts in these topical areas weigh in on this paper. That being said, I appreciate the paper. I think it's really valuable that the authors are pushing the envelope and developing flexible, modular, coupled approaches to model the various contributors to future sea-level rise and their uncertainties.

I have no major issues with the manuscript (though, again, I strongly recommend more expert reviewers weigh in). I have a couple editorial remarks, detailed below. My only one real complaint regards terminology. The authors' use of gravitational, rotational, and deformational (GRD), sterodynamic, and barystatic sea-level contributions (cf. Equation 1) can be inconsistent with definitions in Gregory et al. (2019). I'd rec- ommend the authors either (1.) adopt the definitions used in Gregory et al. (2019) or (2.) acknowledge where their definitions diverge from Gregory et al. (2019) to avoid confusion.

We thank the reviewer for the time spent on the review and for his valuable insights, especially from the Physical Oceanography point of view. We agree with the reviewer's assessment of the advantage of flexible, modular, coupled approaches to model various contributors to future sea-level rise and their uncertainties. We also agree that our initial explanation of the STR and DSL terms were not compatible with the definitions in Gregory et al. (2019), and will definitely tighthen the introduction in this respect. All othere referees pointed to the same issue (see in particular referee #1). We below go through all the comments and try and address them, along with modifications to the manuscript that will be carried out if the editor goes forward with requesting a new version.

Specific comments:

- Line 2: paramount to $->$ important for
  Thank you for the suggestion, we will adopt it in the new manuscript

- Line 4: cost and timing $->$ cost, timing, and risk tolerance

Thank you for the suggestion, we will adopt it in the new manuscript

- Line 23: are summed $->$ are modeled separately and summed
  Thank you for the suggestion, we will adopt it in the new manuscript

- Equation 1. Comparing to Figure 3 in Gregory et al. (2019), I'm confused by this equation. The equivalent equation in Gregory et al. would be: relative sea level = sterodynamic sea level + gravitation, rotation, deformation (GRD) + barystatic - inverted barometer
  In Gregory et al. (2019), GRD includes GIA, and GRD makes no contribution to global mean sea-level changes. What the authors here call "GRD", Gregory et al. (2019) call "contemporary GRD". Anyway, it's fine that the authors here use slightly different terminology. But they should acknowledge where their definitions diverge from Gregory et al. (2019). Otherwise, readers (i.e., I) will get confused.
  We really appreciate the referee checking against Gregory et al. (2019) and seeing this inconsistency. We will definitely remark in the manuscript on the differences in our approach, and refer to our GRD as contemporary GRD. Here is the new paragraph that will be in the manuscript starting at line 37: *Note here that our definition of GRD is not completely in line with Gregory et al. (2019), as GIA is considered as a separate contributor, and the GRD contribution does contribute to global mean sea-level changes. It is rather in line with the definition of contemporary GRD in Gregory et al. (2019).*

- Line 37: local thermosteric $->$ global-mean thermosteric
  This was picked up by all referees, and can lead to confusion about the definition of STR and DSL. Thank you for spotting it, we refer to the referee #1 comments on how we addressed this

- Line 53: qualities $->$ quantities
  Thank you for the suggestion, we will adopt it in the new manuscript

- Line 61: "stays constant in time" it's unclear what the authors mean by this phrase
  We will replace by *is constant through time*

- Line 107: paramount to $->$ important for
  Thank you for the suggestion, we will adopt it in the new manuscript

- Lines 124-126: I'm unfamiliar with studies doing this for projection purposes. Do the authors have a reference in mind for this technique?
  Referre #1 also requested a reference. There is not one involving a projection, but we provided (eg. Thompson et al., 2016, Fig.3) for a good explanation of the approach that could readily be adapted to a projection.

- Line 127: can drive redistribution $->$ can be coupled to redistribution
  Thank you for the suggestion, we will adopt it in the new manuscript

- Line 128: causes $->$ manifests in
  Thank you for the suggestion, we will adopt it in the new manuscript

- Line 129: cause a change in the load of $->$ load
  Thank you for the suggestion, we will adopt it in the new manuscript

- Line 130: SAL effects. Suggest to reference, e.g., Ray (1998), Stepanov and Hughes (2004), and/or Vinogradova et al. (2015) on these points.
  Thank you for the suggestion, we will adopt it in the new manuscript and reference these studies

- Line 131: Please add "made by atmosphere-ocean general circulation models (AOGCMs)" after projections and before the Richter et al. (2013) reference
  Thank you for the suggestion, we will adopt it in the new manuscript

- Line 148: local $->$ global-mean
  Thank you for catching this typo that was also important to all three other referees. We have corrected the manuscript accordingly.

- Line 152-153: The authors should clarify whether they remove the global-mean OBP value or not. If not, are the authors making the Greatbatch correction to account for the Boussinesq nature of most CMIP AOGCMs?
  Thank your for the comment. Indeed we remove the global-mean OBP from ocean models, since ocean dynamics don't add or remove any mass from/to the ocean. The CMIP5 and CMIP6 models all (should) have applied the Greatbatch correction, as confirmed by the fact the global-mean value of the 'zos' fields is zero, and we use the 'zostoga' to the models to get the global thermosteric rise. We will add the following paragraph starting at line 155: *"Note also that the global-mean OBP is removed from the ocean models, since ocean dynamics don't add or remove any mass from/to the ocean. In addition, our projections rely on CMIP5 and CMIP6 fields 'zos' (the sea-surface height change above geoid, or DSL term) and 'zostoga' (global average thermosteric sea-level change or STR) where the Greatbatch correction has been applied, resulting in a zero global-mean value of STR."*

- Line 165: The authors should precisely define the ocean function O(theta,phi) for clarity.
  Thank you for spotting this issue, we will define the ocean function in the manuscript succinctly as *O=1 for oceans and zero otherwise"*

- Line 176: The former $->$ These
  Thank you for the suggestion, we will adopt it in the new manuscript

- Line 193: Please define BAMG on first use
  Thank you for the suggestion, we will define BAMG in the manuscript

- Figure 3 caption: thsi $->$ this
  Thank you for spotting the type, we will correct it in the new manuscript

- Line 246: alphas $->$ alpha is
  Thank you for the suggestion, we will adopt it in the new manuscript

- Line 256-257: "We display ... the average" I don't understand this sentence, but maybe it's just me.
  This sentence can indeed be clearer, we replace with *"We display the average thinning rate $\mu$, $\mu + 3\sigma$ and $\mu - 3\sigma$ (for an arbitrary value of the standard deviation $\sigma = 5\%$)."*

- Line 262: "the KOPP14 ... SLPS framework" Unclear. Are the authors saying that the approach here reduces to and reproduces the Kopp results under certain strong assumptions? Please clarify.
  We are indeed saying that the approach is equivalent to KOPP14 if we use the same partitioning. The assumptions are not so strong, just that the partitioning be the same. However, as demonstrated by Fig. 11, this is not the case anymore once several basins are introduced. We will try and capture this better in the manuscript, with the following statement: *"Once several partitions are adopted however, the refinement in the fingerprint patterns significantly departs from the KOPP14 approach. "*.

- Line 289: please add spaces between -1.65, sigma, to, 1.65, sigma
  Thank you for the suggestion, we will adopt it in the new manuscript

- Line 291-292: "DSL is not ... CMIP5 NorESM-ME runs" Why only one model and why this model? Variance in model projections of DSL changes can be large and important locally.
  In this demonstration of the capabilities of ISSM-SLPS, we wanted to approach the geodetic angle. Significant variance in model projections are indeed found in the CMIP5 and CMIP6 benchmarks, which completely occultate any other variance from any other inputs. We wanted to avoid this. We ask the reviewer to allow for this exception, as we believe it leads to a better validation of our capability.

- Line 303: "Bayesian exploration approach" The authors reference such an approach several times, but never explain it or give a reference.
  We agree with the reviewer. The GIA statistics relied upon here are from Caron et al. (2018), but the bayesian framework we refer to is described in Caron et al. (2017) and is based on a bayesian inversion method using Simulated Annealing (Kirkpatrick et al., 1983), a variation of the Monte Carlo with Markov chains (MCMC) method (Metropolis and Ulam, 1949; Metropolis et al., 1953). We will better refine the description in the manuscript and give extended citations. The paragraph will now read *These statistics were evaluated using bayesian inversion method based on Simulated Annealing (Kirkpatrick et al., 1983), a variation of the Monte Carlo with Markov chains (MCMC) method (Metropolis and Ulam, 1949;*

*Metropolis et al., 1953). They can be used directly in SLPS, either during a standard probabilistic projection run, or a posteriori as is the case here. These statistics reflect the statistical fitness to a global GIA dataset composed of paleo-RSL indicators and vertical GPS trends.*

- Line 311: "the tails are much larger for the 1 basin scenario" This seems like a very important results, but I don't think the authors have discussed it enough for me to understand physically why this is the case. Suggest to consider adding more of a description.

  We agree with the reviewer. The reason for reduced tails is that by multiplying the number of basins, we recompute fingerprints that are more reflective of the true spatial pattern. The example of New York used in Larour et al. (2017) helps in understandig this feature: the entire South-East Greenland contributes zero sea-level change in NY. If a basin is positioned over this entire region, it will contribute zero variance to the PDF distribution for SLR in NY. This leads to a reduction in the tails of the distribution. We will add this explanation in the manuscript too, as suggested. Here is the text we will add starting at line 317: *" This can be visualized better by taking the example of New York, where following Larour et al. (2017) contributions from South Greenland are almost negligible. This implies that all the basins (and corresponding GRD patterns) in South Greenland will contribute zero variance to the PDF for RSL at New York. This will therefore result in smaller tails for projections that rely on more refined basins."*

- Line 311: the "likely" (5-95%) range − > the width of the "likely" (5-95%) range

  Thank you for the suggestion, we will adopt it in the new manuscript

- Line 325: siginificantly − > significantly

  Thank you for spotting the typo, we will correct the manuscript accordingly

- Line 343: urther − > further

  Thank you for spotting the typo, we will correct the manuscript accordingly

**References**

Caron, L., Métivier, L., Greff-Lefftz, M., Fleitout, L., and Rouby, H.: Inverting Glacial Isostatic Adjustment signal using Bayesian framework and two linearly relaxing rheologies, Geophysical Journal International, 209, 1126–1147, 2017.

Caron, L., Ivins, E. R., Larour, E., Adhikari, S., Nilsson, J., and Blewitt, G.: GIA Model Statistics for GRACE Hydrology, Cryosphere, and Ocean Science, Geophysical research letters, 45, 2203–2212, 2018.

Gregory, J., Griffies, S., Hughes, C., et al.: Concepts and Terminology for Sea Level: Mean, Variability and Change, Both Local and Global, Surv. Geophys., 40, 1251–1289, 2019.

Kirkpatrick, S., Gelatt, C. D., and Vecchi, M. P.: Optimization by Simulated Annealing, Science, 220, 671–680, https://doi.org/10.1126/science.220.4598.671, 1983.

Larour, E., Ivins, E. R., and Adhikari, S.: Should coastal planners have concern over where land ice is melting?, Science Advances, 3, e1700 537, 2017.

Metropolis, N. and Ulam, S.: The Monte Carlo method, J. Amer. Stat. Associ., 44, 335–341, 1949.

Metropolis, N., Rosenbluth, A. W., Rosenbluth, M. N., Teller, A. H., and Teller, E.: Equation of State Calculations by Fast Computing Machines, The Journal of Chemical Physics, 21, 1087–1092, https://doi.org/10.1063/1.1699114, 1953.

Thompson, P. R., Hamlington, B. D., Landerer, F. W., and Adhikari, S.: Are long tide gauge records in the wrong place to measure global mean sea level rise?, Geophysical research letters, 2016.

---

## Author Comment (AC2) · 28 Jul 2020

**1 Response to Anonymous Referee #2**

The authors developed a new method to account for feedbacks between ice sheets, ocean circulation and solid-Earth deformation through dynamically computed sea-level fingerprints for probabilistic projections of local sea-level change. This is a significant and timely contribution because current probabilistic projections use static fingerprints and cannot readily incorporate advances in forward models. The manuscript is well- structured and provides clear examples of how the SLPS works. However, I think the clarity of the manuscript could be improved with some (minor) modifications of the text and figures, see the comments below.

We thank the referee for the time spent reviewing the manuscript and for the positive assessment of the manuscript, in particular the capability of ISSM-SLPS to readily update projections with new forward modeling advances that are tightly coupled with the framework. We address below all the concerns from the referee, and present future changes to the manuscript that will be implemented if the editor moves forwards with accepting corrections.

- L4: For long-term projections (. . .) that provide such probabilistic projectionstive, can this be rewritten?
  Indeed this is too heavy, we will reformulate to *"For a time horizon of 100 years, frameworks have been developed that provide such projections by relying on ..."*

- L8: solid-Earth 'uplift' – displacement or deformation would be more complete?
  We thank the reviewer for the comment, but believe that solid-Earth uplift is a terminology used throughout the Cryosphere/solid-Earth community that has a very specific meaning. Deformation could convey lateral motion too, as well as displacement. Our focus here is on deformation that impacts RSL through VLM in particular. We respectfully would like to keep the terminology as is.

- L26: why not give examples of the use of projections globally rather than so many references for just the US?
  We understand the concern of the reviewer, however, here the intent of the manuscript was to point out to how widely the KOPP14 framework has been adopted. For assessment outside the US, the onus is significantly more on the IPCC assessments, which are arguably, given their 4 year cycle, not as responsive to more recent developments in the science community. Given that our intent was focused on the KOPP14 framework we would like to respectfully request to keep ours references as is.

- L37 & 118: sterodynamic sea-level change is defined incorrectly here, see Gregory et al. (2019). It is the sum of global (not local) thermosteric expansion and ocean dynamics (which include the local steric effect, both thermo- and halosteric) including the IB effect. Can the authors clarify

this?

Another referee #1 pointed to the same issue, and we thank you for spotting this issue. The definition for STR and DSL for our manuscript indeed follows Gregory et al. (2019). The confusion came from the erroneous use of "local" instead of "global" at line 37. The sentence will now read *"which is the sum of globally averaged thermosteric expansion and local sea-level changes due to ocean dynamics (which include the local steric effect, both thermosteric and halosteric)."*

- L38 & 55: Not just ESMs but also AOGCMs
  Duely noted, and the manuscript will be updated accordingly

- L47: Do you mean Kopp et al. (2017) here instead of (2014)?
  Indeed that is what we meant, thank you for spotting the typo, will be reflected in the amended manuscript.

- L64-66: Can the authors comment on the expected importance of geometry changes to 21st century sea-level projections?
  Referee #1 had a similar comment referring to the fact that we did not demonstrate the importance of time variable fingerprints. We will add an example figure (Fig. 1) of the evolution of $RSL_{GRD}$ for the Greenland Ice Sheet, using a projection from ISMIP6 (Goelzer et al., 2020, accepted) based on the ISSM JPL run for experiment 5. The nominal fingerprints are shown to be significantly different between 2015 and 2100. We refer the reader to the response to referee #1 for the figure, comments, and corresponding changes to the manuscript that address the present comment also.

- L83: 'results such as (. . .) results' - please rewrite
  We will rewrite this sentence to *"In order to be able to account for strong couplings, or to even be able to ingest recent modeling results, one needs to propagate the local mass changes and the associated uncertainties into regional sea-level projections. This is particularly relevant now given new modeling runs that have been carried out within large Modeling Intercomparison Projects (MIPs) such CMIP5 and CMIP6, as well as ISMIP6 or GlacierMIP2."*

- L91: 'higher-frequency' higher than what? Perhaps use 'high-frequency' instead?
  Thank you for the suggestion. We will replace to "high-frequency", and better define what is meant, with frequencies of interest being daily to monthly.

- L97-103: this paragraph misses a final sentence placing its content in context of the manuscript
  We agree with the reviewer, and will add the following sentence at the end of the paragraph *"All these advances need to be fully integrated into new*

*probabilistic projections of sea-level change, and a new approach therefore needs to be envisioned that will allow for such new processes to be accurately modeled."*

- L104-107: this sentence doesn't read very well, can you please split this up in smaller sentences?
  We agree with the reviewer. Actually, the whole paragraph will be recast to *" Indeed, moving from strategies where continental scale mass changes are sampled and multiplied with the corresponding fingerprint, to actually sampling upstream model inputs is paramount to improving the state of the art. In particular, there is a strong need to fully account for spatial patterns of mass change and their uncertainty (see e.g. Fig. 1b-d), This applies to among others SMB, basal friction, or ice and solid-Earth rheological properties."*

- L114-115: 'and through . . . individual sample' misses a verb, perhaps reverse the order with 'retaining the. . .from Eq. 1'
  We will rephrase the paragraph to *" We improve the existing process-based approach by using the Ice-Sheet and Sea-Level System Model (ISSM, Larour et al, 2012c) which allows for inclusion of forward model physics. It also improves the modeling and sampling of covariances between input processes, both temporally and spatially through the computation of high-resolution barystatic-GRD patterns. "*

- Figure 3: the axis labels have fallen off the figure.
  Thank you for spotting this issue, will be fixed in the amended manuscript

- Figure 5: It is very hard to distinguish colors of cells within the mesh, could this be improved by for example adapting the range of the colorbar?
  We thank the reviewer for the suggestion. By saturating the colorbar, we reach better contrast. We will do so for the manuscript, and provide an improved figure. We will also amend the caption to explain why we saturated the colorbar.

- L253: has a typo 'appraoch'
  Thank you for spotting the typo, will be corrected in the manuscript

- L291-292: "DSL is not sampled, but rather deterministically set to the DSL term of the CMIP5 NorESM-ME runs" – why? please explain
  This is due to the uncertainty in the quality of the CMIP5 runs in terms of global mean thermosteric contribution. We preferred to avoid this uncertainty and concentrate on the geodetically relevant components given the scope of the manuscript.

- L299: refers to figure 9 instead of 8, and discussion of Figure 9 is missing?
  We refer to referee #1 comments, which also spotted this issue.

- L316: "inpact"
  Thank you for the typo, will be corrected in the manuscript.

- L343: 'urther'
  Thank you for spotting the typo, will be corrected in the manuscript.:w

**References**

Goelzer, H., Nowicki, S., Payne, A., Larour, E., Seroussi, H., Lipscomb, W. H., Gregory, J., Abe-Ouchi, A., Shepherd, A., Simon, E., Agosta, C., Alexander, P., Aschwanden, A., Barthel, A., Calov, R., Chambers, C., Choi, Y., Cuzzone, J., Dumas, C., Edwards, T., Felikson, D., Fettweis, X., Golledge, N. R., Greve, R., Humbert, A., Huybrechts, P., Le clec'h, S., Lee, V., Leguy, G., Little, C., Lowry, D. P., Morlighem, M., Nias, I., Quiquet, A., Rückamp, M., Schlegel, N.-J., Slater, D., Smith, R., Straneo, F., Tarasov, L., van de Wal, R., and van den Broeke, M.: The future sea-level contribution of the Greenland ice sheet: a multi-model ensemble study of ISMIP6, The Cryosphere, https://doi.org/10.5194/tc-2019-319, URL https://www.the-cryosphere-discuss.net/tc-2019-319/, 2020, accepted.

Gregory, J., Griffies, S., Hughes, C., et al.: Concepts and Terminology for Sea Level: Mean, Variability and Change, Both Local and Global, Surv. Geophys., 40, 1251–1289, 2019.

---

## Author Comment (AC3) · 28 Jul 2020

**1 Response to Anonymous Referee #1**

The authors present a framework to produce probabilistic sea-level projections which allows to incorporate and update contributions from different models of the individual drivers in a flexible, yet consistent, way. It builds on the ISSM-SESAW sea level solver by Adhikari et al. (2016), which was also presented in this journal and which is based on an unstructured mesh, allowing high accuracy at very limited computational costs. The manuscript is clear, it reads well and it presents enough graphics to show the working and capabilities of the software package. Below, I am suggesting a number of minor improvements, listed in the order as they appear in the manuscript.

We thank the referee for the time spent reviewing the manuscript, and for the positive assessment of the manuscript and the methodology implemented in the ISSM-SLPS projection capability. We have tried to address all the concerns raised, as well as present the changes that will be implemented in the amended manuscript to be submitted once the editor requests.

- Line 60: The argument about the time invariancy of fingerprints does not seem very strong, since it is always possible to increase the number of fingerprints for a given source, with the option of using independent scaling factors for each sub-source. Besides, the authors do not seem to demonstrate the effect of time-variable fingerprints on sea level projections. The improvement could be from a computational point of view (due to, e.g., the modularity of the approach), but then it could be explained more clearly.

We agree with the referee that it is always possible to increase the number of fingerprints for a given source, and of using several across time too. However, doing so converges towards a solution that we effectively generalized in our framework, where we can handle any type of thickness change pattern at whichever resolution is required, and at whatever time resolution is optimal. We agree with the reviewer that we have not in the manuscript shown the effect of time variable fingerprints explicitly, though we did compute temporally variable fingerprints to generate Fig. 8. Fig. 1 of the present document shows the impact of temporarlly variable fingerprints using an ISMIP6 model projection of ice thickness changes in Greenland. The figure clearly demonstates the significant impact between normalized RSL fingerprints at 2016 vs 2045 and 2075 , with significantly different spatial patterns down to Marocco or Cuba, and over the entire arctic. We will add this figure to the manuscript, as it quantifies clearly the need for temporally variable fingerprints, and we will add the following text in the manuscript:
*An example of the inadequacy of temporally constant fingerprints is shown in Fig. 1 for a projection of Greenland's contribution to RSL at 2016 vs 2045 and 2075. Normalized RSL patterns are clearly different between 2016 and 2045, and the differences are not just local to Greenland, but spill over into North Europe, Alaska, the Canadian arctic, etc...*

- Line 65: Not sure what "profound geometry" means, it is possibly a typo.
We will replace "profound geometry" with "pronounced spatial pattern", which was the original intended meaning.

- Line 124: Could you add a reference about the "alternative approach" described here?
Thank you for the suggestion. We will add Thompson et al. (2016) (Fig.3) as an example of this approach.

- Line 148: I guess the term STR refers to a global mean value, not a local one as stated, since $RSL_{STR}$ only depends on time.
We thank the reviewer for catching this, indeed STR refers to the global mean value. We will correct the manuscript accordingly.

- Line 168: I am surprised by the fact that $RSL_{GRD}$ includes viscoelastic deformation of the solid earth, rather than just elastic as in most studies of this kind. If this is the case, it should be highlighted in the introduction.
We thank the reviewer for spotting this. Indeed it should be explicitly noted that visco-elastic deformation is currently allowed in the projection system. Here, we define visco-elastic deformation any deformation that is not related to GIA, but to smaller time-scale loading processes (50 to 100 years) as observed in West Antarctica Barletta et al. (2018). In practice, this type of deformation is not yet handled in ISSM, but will in the future. We will make the following explicit comment in the manuscript at line 151: *This implies that viscoelastic deformation is split between long-term time scales and short-term fast rebound of the bedrock uplift, such as observed in West Antarctica (Barletta et al., 2018), acting essentially oveer time scales of 50-100 years.*

- Line 215: I suggest adding that the number of elements is significantly smaller than in the case of an equi-angular 1x1-deg grid, which would already be rather coarse.
We thank the reviewer for the suggestion, indeed the equi-angular grid at 1x1 deg would require 64,800 vertices, with our mesh standing at only 19,486. We will add this to the manuscript.

- Lines 225-241: I am not sure whether showing a few lines of code is normal in GMD, but, if possible, it would be nice to replace this by a flow chart.
We understand the reviewer's point of view. We have debated this extensively, having in the past presented pieces of code in GMD. However, this is the main algorithm core, and we believe it should be explicitly and exactly described. The goal here is to avoid any confusion that could arise from a flowchart. We respectfully ask to keep the code as is, but will defer to the editor's decision.

- Line 299: It seems a few words are missing, since Fig.8b shows GMSL values, while values for 9 cities around the world are shown in Fig.9. In addition, the caption of Fig.9 does not mention the timeframe for which the projections are generated.
We thank the reviewer for pointing this redirection to the wrong figure. We

will correct the manuscript accordingly, and add a quick description of Fig 8b in the text as showing the evolution of GMSL's PDF through time. We will also add the time frame in Fig. 9.

- Line 304: I suggest also mentioning that the width of the two PDFs shown in Fig.10 is different, rather than only the mean, because it makes an even stronger point about the benefit of this approach.
  We thank the reviewer for the suggestion. After checking though, the PDFs standard deviations are only different by .4% on a relative basis. The PDFs do indeed appear different at their basis, but we believe this is a visual effect from the color choice. We will put a comment to this effect in the caption of Fig. 10 to dispel any confusion.

- Line 316: "impact" (spelling).
  Thank you for pointing out the typo, we will correct the manuscript accordingly.

- Line 338: The importance of "geodetically compliant patterns", in spite of being part of the manuscript title, is nowhere really discussed nor explained. From this line, my understanding is that it refers to the possibility of defining the mesh in such a way that it matches the location of specific geodetic observations, hence avoiding unnecessary interpolations. In any case, the issue warrants a more extensive discussion, possibly in the introduction.
  We thank the reviewer for the comment. Geodetically compliant referred to the fact that ISSM-SLPS computes spatio-temporal GRD patterns throughout the projection sampling, which is not always done consistently for other similar frameworks. We will add comments in the introduction referencing this explanation explicitly. In particular, we will add the following at line 117: *"The latter feature builds the basis for a geodetically compliant projection system where GRD patterns and their computations is done systematically and does not introduce biases in the RSL projections."*

**References**

Barletta, V. R., Bevis, M., Smith, B. E., Wilson, T., Brown, A., Bordoni, A., Willis, M., Khan, S. A., Rovira-Navarro, M., Dalziel, I., Smalley, R., Kendrick, E., Konfal, S., Caccamise, D. J., Aster, R. C., Nyblade, A., and Wiens, D. A.: Observed rapid bedrock uplift in Amundsen Sea Embayment promotes ice-sheet stability, Science, 360, 1335–1339, 2018.

Goelzer, H., Nowicki, S., Payne, A., Larour, E., Seroussi, H., Lipscomb, W. H., Gregory, J., Abe-Ouchi, A., Shepherd, A., Simon, E., Agosta, C., Alexander, P., Aschwanden, A., Barthel, A., Calov, R., Chambers, C., Choi, Y., Cuzzone, J., Dumas, C., Edwards, T., Felikson, D., Fettweis, X., Golledge, N. R., Greve, R., Humbert, A., Huybrechts, P., Le clec'h, S., Lee, V., Leguy, G., Little, C., Lowry, D. P., Morlighem, M., Nias, I., Quiquet, A., Rückamp, M., Schlegel, N.-J., Slater, D., Smith, R., Straneo, F., Tarasov, L., van de Wal, R., and van den Broeke, M.: The future sea-level contribution of the Greenland ice sheet: a multi-model ensemble study of ISMIP6, The Cryosphere, https://doi.org/10.5194/tc-2019-319, URL https://www.the-cryosphere-discuss.net/tc-2019-319/, 2020, accepted.

Thompson, P. R., Hamlington, B. D., Landerer, F. W., and Adhikari, S.: Are long tide gauge records in the wrong place to measure global mean sea level rise?, Geophysical research letters, 2016.

[Figure]

[Figure]

[Figure]

Figure 1: Normalized fingerprints for Greenland at 2016, 2045 and 2075, based on the JPL ISSM experiment 5 simulation contributed to ISMIP6 (Goelzer et al., 2020, accepted). Variations in ice-thickness change patterns are significantly different between the East and West coast of Greenland, and along a South-West gradient too, resulting in significantly different contributions to local RSL at different time snapshots

---

## Author Response (AR2)

**1 Letter to Editor: minor revisions**

Dear Editor,

please find below our response to the questions/comments from the reviewers for this minor revision, along with the new manuscript and an annotated version of the manuscript showing the changes. We thank you for the time spent on the manuscript and for guiding it towards publication. It is now much improved in our opinion, and we are greatful for all the help pushing it to this stage.

Best regards,

Eric Larour and co-authors.

- Line 35 and 145: The acronym "GLA" is defined in both places. I think the first of these definitions is unnecessary? We agree with the reviewers, and we took out the GLA definition at line 35.

- Line 36: The acronym GIA is first referenced here, although it is not defined until line 42. We thank the reviewers for spotting this issue. We definte GIA at line 36 now

- Lines 41-42: Please explicitly state the name of the term $RSL_{GIA}(\theta, \phi, t)$ in the text, as you do when discussing each of the other terms in Equation 1. Thank you for the suggestion, we will explicitly state the name of the term $RSL_{GIA}$ in the text, to be consistent.

- Line 69: Replace "vs" with "versus". Done.

- Lines 69-70: The latter part of this sentence could be better written as "... into regions such as North Europe, Alaska and the Canadian Arctic". This description of the figure could also be longer and more detailed. For example, the figure shows results for 2075 but these are not discussed. Please either include 2075 in your description, or consider removing it from the figure. We replaced the latter part of the sentence as advised, and indeed expanded the figure caption, and referenced the 2075 time explicitely in the text.

- Line 102: Replace "like" with "such as". Done.

- Line 148: Remove "are" after "fingerprints". Done.

- Line 164: Replace "oveer" with "over". Done. Thanks for finding the typo.

- Lines 181-182: I think "over the oceans" would be better than "on oceans"? Thank you for the suggestion, we replaced accordingly in the manuscript.

- Line 187: Insert "the" before "geoid". Done.

- Line 234: Change the end of the sentence to "which is three times as many as for a coarse grid resolution" or similar. Thanks for the suggestion, done.

- Figure 1: I find the second sentence of the caption confusing. Are you referring to variations in space, in time, or both? A longer and more descriptive caption would be desirable. Thank you for the suggestion, we have improved the caption, and dispelled the confusion about variations. What we meant is that the spatial patterns were different at all three epochs.

- Figure 3: In the caption, insert "of" after "Diagram". Done.

[revised manuscript text omitted]

---

## Author Response (AR3)

**1 Letter to Editor: minor revisions**

Dear Editor,

please find attached our final correct version of the manuscript. We have replaced the caption for Figure 1 with your suggested caption. Thank you again for leading the manuscript through the whole review process.

Kind regards,

Eric Larour and co-authors.

[revised manuscript text omitted]